EMBO
Molecular Medicine

# Very-low-carbohydrate diet enhances human T-cell immunity through immunometabolic reprogramming

Simon Hirschberger[1,2,†] (ID), Gabriele Strauß[1,2,†], David Effinger[1,2], Xaver Marstaller[1], Alicia Ferstl[1] (ID), Martin B Müller[1,2], Tingting Wu[1], Max Hübner[1,2], Tim Rahmel[3], Hannah Mascolo[1], Nicole Exner[4], Julia Heß[5,6], Friedrich W Kreth[1], Kristian Unger[5,6] & Simone Kreth[1,2,*] (ID)

## Abstract

Very-low-carbohydrate diet triggers the endogenous production of ketone bodies as alternative energy substrates. There are as yet unproven assumptions that ketone bodies positively affect human immunity. We have investigated this topic in an *in vitro* model using primary human T cells and in an immuno-nutritional intervention study enrolling healthy volunteers. We show that ketone bodies profoundly impact human T-cell responses. CD4[+], CD8[+], and regulatory T-cell capacity were markedly enhanced, and T memory cell formation was augmented. RNAseq and functional metabolic analyses revealed a fundamental immunometabolic reprogramming in response to ketones favoring mitochondrial oxidative metabolism. This confers superior respiratory reserve, cellular energy supply, and reactive oxygen species signaling. Our data suggest a very-low-carbohydrate diet as a clinical tool to improve human T-cell immunity. Rethinking the value of nutrition and dietary interventions in modern medicine is required.

**Keywords** immunometabolism; ketogenic diet; metabolic therapy; nutritional intervention; T-cell immunity
**Subject Categories** Immunology; Metabolism

## Introduction

Western diet is increasingly seen as being at the root of many diseases, such as metabolic syndrome, autoimmune disorders, and cancer, and thus is suspected to limit life expectancy during the 21st century (Olshansky *et al*, 2005; Christ & Latz, 2019). It impairs cellular immunity and evokes systemic low-grade inflammation not only by causing obesity but also by direct reprogramming of immune cells toward a proinflammatory phenotype (Hotamisligil *et al*, 1993; Visser *et al*, 1999; Weisberg *et al*, 2003; Mathis, 2013; Guo *et al*, 2015; de Torre-Minguela *et al*, 2017; Dror *et al*, 2017; Christ *et al*, 2018). Nutritional interventions may hold promise as a tool to prevent and even to treat disease. Unfortunately, most recommendations on food intake and dietary guidelines yet lack substantiated scientific background (Archer *et al*, 2018; Ioannidis, 2018; Archer & Lavie, 2019; Ludwig *et al*, 2019). Novel nutritional concepts promote a restriction of carbohydrates in favor of fat to ameliorate detrimental low-grade inflammation (Paoli *et al*, 2015; Bosco *et al*, 2018; Myette-Côté *et al*, 2018). However, large observational studies investigating this approach are highly controversial (Dehghan *et al*, 2018; Seidelmann *et al*, 2018), and molecular data in humans are scarce.

In this regard, the high-fat low-carbohydrate ketogenic diet (KD) is one highly discussed approach (Bolla *et al*, 2019; Ruiz Herrero *et al*, 2020). Restriction of carbohydrate intake leads to the endogenous production of ketone bodies such as beta-hydroxybutyrate (BHB) as evolutionary conserved alternative metabolic substrates, which can be utilized via mitochondrial oxidative phosphorylation (Puchalska & Crawford, 2017). In animal models, BHB has been shown to dampen inappropriate innate immune responses via suppression of the NLRP3 inflammasome, thus ameliorating chronic low-grade inflammation and associated diseases (Youm *et al*, 2015; Goldberg *et al*, 2017; Newman *et al*, 2017). Human adaptive immunity, however, has not yet been addressed (Stubbs *et al*, 2020).

Here, we present the first study investigating the influence of KD on human immune responses *in vitro* and in a cohort of healthy subjects. Our results reveal profound beneficial effects of ketone

1 Walter Brendel Center of Experimental Medicine, Ludwig-Maximilian-University München (LMU), Munich, Germany
2 Department of Anaesthesiology and Intensive Care Medicine, Research Unit Molecular Medicine, LMU University Hospital, Ludwig-Maximilian-University München (LMU), Munich, Germany
3 Department of Anesthesia, Intensive Care Medicine and Pain Therapy, University Hospital Knappschaftskrankenhaus Bochum, Bochum, Germany
4 Metabolic Biochemistry, Biomedical Center (BMC), Faculty of Medicine, Ludwig-Maximilian-University München (LMU), Munich, Germany
5 Helmholtz Center Munich, Research Unit Radiation Cytogenetics, Neuherberg, Germany
6 Department of Radiation Oncology, LMU University Hospital, Ludwig-Maximilian-University München (LMU), Munich, Germany
*Corresponding author. Tel: +49 89 2180 76508; E-mail: simone.kreth@med.uni-muenchen.de
†These authors contributed equally to this work

bodies on human T-cell immunity. BHB improves effector and regulatory T-cell function and primes human T memory cell differentiation both *in vitro* and *in vivo*. These functional changes are based on a fundamental immunometabolic reprogramming, resulting in enhanced mitochondrial oxidative metabolism, thus conferring an increased immunometabolic capacity to human T cells. We provide molecular evidence that ketone bodies promptly improve human T-cell metabolism and immunity. By complementing classical approaches of modern medicine, nutritional interventions offer new perspectives for prevention and therapy of numerous diseases.

# Results

### BHB improves human T-cell immune capacity

To evaluate the impact of ketone bodies on human immunity, human peripheral blood mononuclear cells (PBMCs) were cultivated for 48 h with D/L-BHB (BHB$^+$) under stimulating conditions. In dose-finding experiments, only 10 mM BHB significantly enhanced human Pan T-cell immune response (Appendix Fig S1A and B). 10 mM racemic BHB corresponds to approximately 5 mM metabolically active D-BHB, resembling a near-maximum level of endogenous ketone production (Puchalska & Crawford, 2017). Thus, 10 mM D/L-BHB was used to further decipher the impact of short-term BHB incubation on human T-cell immunity. BHB led to a significant transcriptional upregulation of CD4$^+$ T-cell cytokines interleukin (IL)2, IL4, IL8, and IL22 (Fig 1A). Protein analyses also showed an upregulation of IL2, IL4, IL6, and IL8 (Fig 1B). CD4$^+$ subset analysis revealed a downregulation of the Th1 transcription factor Tbet—resulting in a decrease in Tbet/GATA3 ratio—yet no significant changes of the Th$_1$/Th$_2$ cell ratio (Fig 1C, Appendix Fig S1E). CD8$^+$ T-cell response displayed markedly increased levels of interferon γ (IFNγ), cytolytic proteins perforin 1 (PRF1) and granzyme B (GZMB), cytotoxic T lymphocyte-associated antigen 4 (CTLA4), and tumor necrosis factor alpha (TNFα; Fig 1D). IFNγ and TNFα protein secretion was also elevated in response to BHB, and functional analysis unveiled enhanced cell lysis activity (Fig 1E). Of note, in the absence of activating stimuli, unstimulated T cells did not exhibit relevant expression levels of immune cytokines, irrespective of their nutritional situation (Appendix Fig S1C and D).

Regulatory T cells (T$_{reg}$) are indispensable to control T-cell activity. T$_{reg}$ differentiation of BHB$^+$ T cells resulted in increased Foxp3 transcription and elevated fractions of CD4$^+$CD25$^+$Foxp3$^+$ T$_{reg}$ cells (Fig 1F). Consequently, the expression of T$_{reg}$ master cytokines TGFβ1 and IL10 was also increased (Fig 1G).

Notably, BHB had no significant impact on functions of the innate immune system. We conducted functional analyses of human whole blood supplemented with 10 mM BHB. Both phagocytic capacity and cellular respiratory burst displayed no change due to BHB (Appendix Fig S1F and G). Furthermore, innate master cytokine IL1β was not significantly upregulated in BHB$^+$ PBMC (Appendix Fig S1H).

In summary, these findings clearly provide evidence for both improved effector and improved regulatory T-cell function, representing an enhanced global immune capacity of BHB$^+$ primary human T cells.

### BHB increases mitochondrial aerobic oxidative metabolism in primary human T cells

We next hypothesized that an elevated energy supply may evoke these effects. To analyze the functional impact of BHB on T-cell metabolism, we performed extracellular flux analyses. BHB-cultivated primary human T cells showed an increased maximal respiration and spare respiratory capacity (Fig 2A). BHB$^+$ CD4$^+$ T cells exhibited elevated basal oxidative respiration (Fig 2B). In CD8$^+$ T cells, a marked shift toward oxidative metabolism—consisting of elevated basal and maximal respiratory rates, mitochondrial ATP production, and spare respiratory capacity—was displayed (Figs 2C and EV1A). Notably, these alterations were not based on suppressed glycolysis, as BHB$^+$ T cells retained their glycolytic capacity (Fig EV1B and C). Taken together, these data provide evidence for a substantial metabolic shift of T cells toward mitochondrial oxidative phosphorylation (OXPHOS) in response to BHB.

### BHB amplifies ROS production and directs T cells toward memory cell formation

The mitochondrial respiratory chain complexes I, II, and III are viewed as the major physiological sources of cellular and

---

**Figure 1. Beta-hydroxybutyrate enhances human T-cell immune capacity *in vitro*.**

Human peripheral blood mononuclear cells (PBMCs) were cultivated for 48 h in RPMI containing 80 mg/dl glucose (NC) and supplemented with 10 mM beta-hydroxybutyrate (BHB). T-cell stimulation was performed through CD3/CD28 Dynabeads at a bead:cell ratio of 1:8. Human pan T-cell RNA was isolated, and cell culture supernatant was sampled.

A   mRNA expression of CD4$^+$ cytokines IL2, IL4, IL8, and IL22 relative to endogenous controls, $n$ = 13/11/10/8 biological replicates.

B   Protein expression of IL2, IL4, IL6, and IL8, analyzed in the supernatant of stimulated PBMCs, $n$ = 9/12/11/11 biological replicates.

C   Th$_1$/Th$_2$ cell transcription factors Tbet and GATA3 quantified via RT–qPCR and flow cytometric ratio of Th$_1$/Th$_2$ cells following differentiation, $n$ = 6/6/4 biological replicates.

D   mRNA expression of CD8$^+$ cytokines IFNγ, PRF1, TNFα, GZMB, and CTLA4 in stimulated human T cells relative to internal controls, $n$ = 15/14/10/14/9 biological replicates.

E   Protein expression of IFNγ/TNFα in the supernatant of stimulated PBMCs and cell lysis-dependent calcein fluorescence of isolated T cells, $n$ = 8/12/9 biological replicates.

F   Foxp3 mRNA relative to internal control in human PBMCs and quantification of CD4$^+$CD25$^+$Foxp3$^+$ regulatory T cells (T$_{reg}$) following 5 days of T$_{reg}$ differentiation with a representative Foxp3 histogram plot (right side), $n$ = 7/11 biological replicates.

G   IL10 and TGFβ1 mRNA and protein expression of human T$_{reg}$ cells, $n$ = 8/9 (IL10), 5/9 (TGFβ1) biological replicates.

Data information: Data depicted as mean ± SEM (protein data) and box plots with median, 25$^{th}$ and 75$^{th}$ percentiles and range (all other). Dots indicating individual values. *$P$ < 0.05, **$P$ < 0.01, ***$P$ < 0.001, paired $t$-test/Wilcoxon matched-pairs signed rank test, as appropriate.

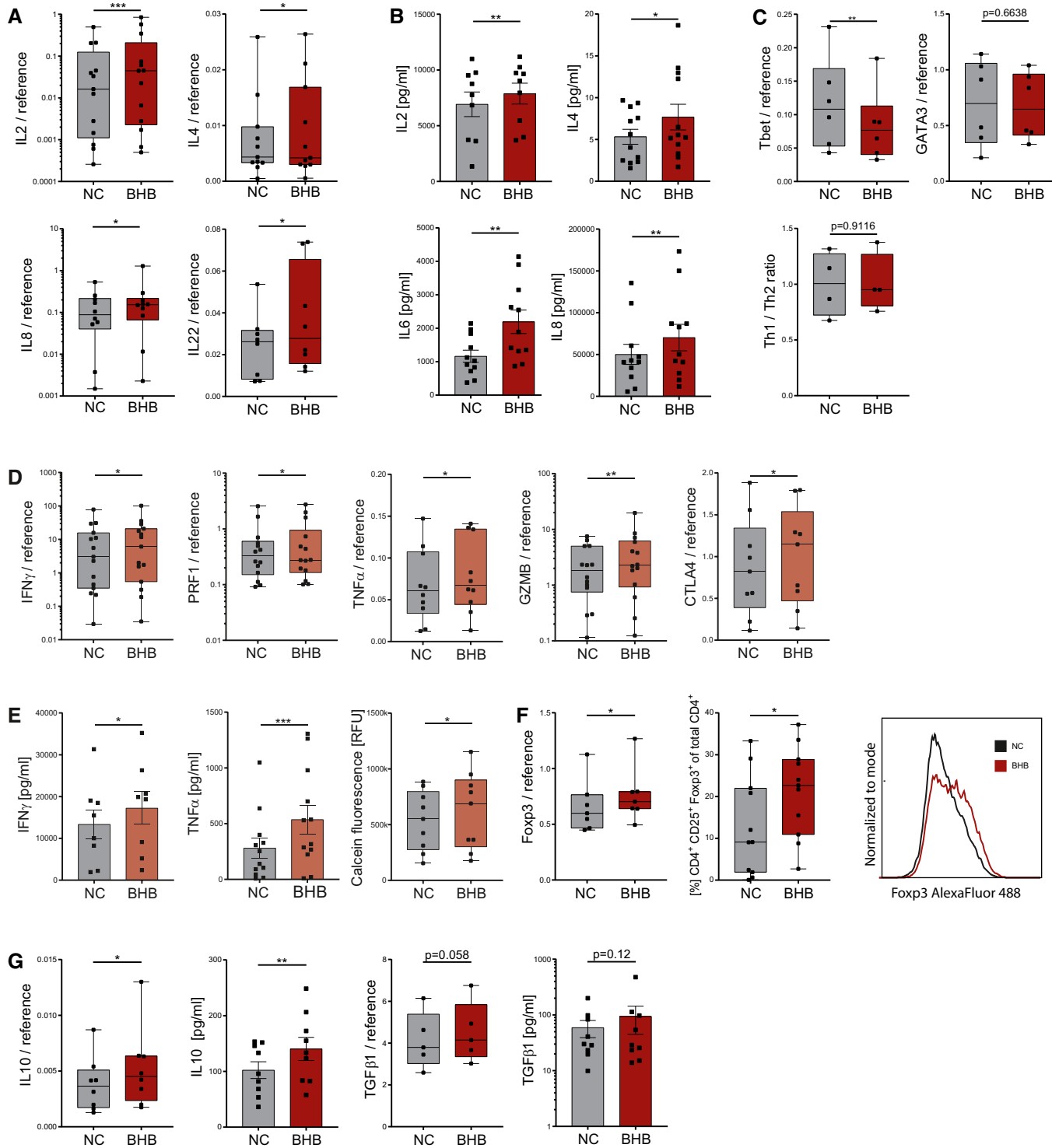

**Figure 1.**

mitochondrial reactive oxygen species ([m]ROS) (Liemburg-Apers *et al*, 2015), and moderate mROS levels have been shown to be required for T-cell activation (Kamiński *et al*, 2012; Sena *et al*, 2013). Therefore, an increase in tricarboxylic acid cycle/electron transport chain (TCA/ETC) activity due to an elevated supply of BHB may result in augmented production of reactive oxygen species, serving as a T-cell "second messenger", hence amplifying T-cell immune capacity. To test this hypothesis, we evaluated cellular and mitochondrial ROS in T cells treated with BHB. Flow cytometry analysis revealed higher cellular ROS in BHB[+] stimulated T cells, which could also be detected in CD4[+] and CD8[+] T-cell subsets (Figs 3A and EV2D). Additionally, in all cell subsets, elevated levels

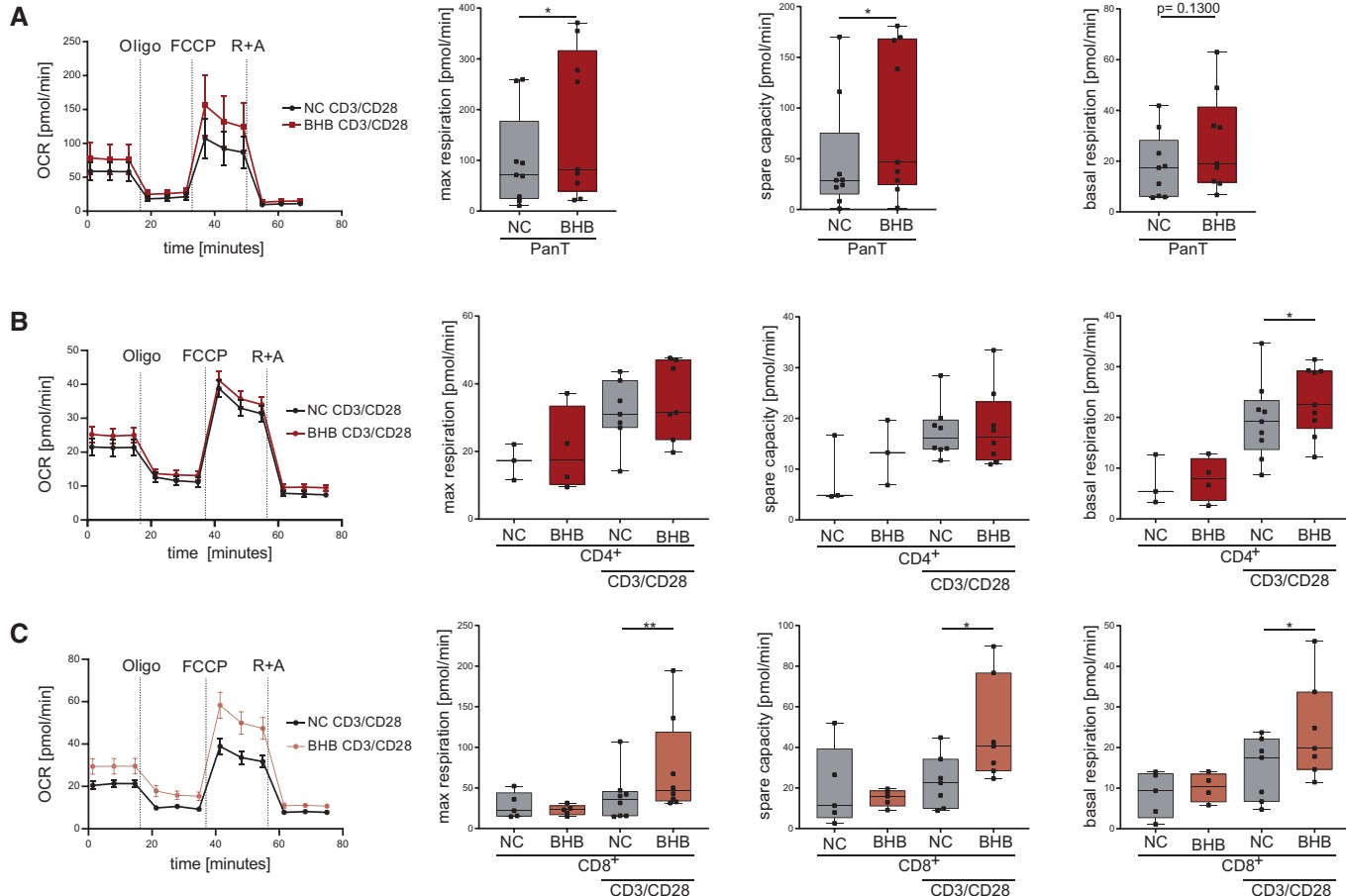

**Figure 2. Ketone bodies shift T-cell metabolism to oxidative phosphorylation.**

Human PBMCs were cultivated for 48 h in RPMI containing 80 mg/dl glucose (NC) and supplemented with 10 mM beta-hydroxybutyrate (BHB). T-cell stimulation was performed through CD3/CD28 Dynabeads at a bead:cell ratio of 1:8. Pan T cells and CD4$^+$ and CD8$^+$ T cells were isolated through magnetic cell separation. Mitochondrial metabolism was analyzed for each subpopulation using a Seahorse XF96 Analyzer.

A–C   OCR, maximum respiration, spare respiratory capacity, and basal respiration were measured in (A) pan T cells, (B) CD4$^+$ T cells, and (C) CD8$^+$ T cells, $n = 9$ (pan T), $n = 3/4$ (CD4 unstimulated), $n = 7/8/9$ (CD4 stimulated), $n = 5$ (CD8 unstimulated), and $n = 8/7/7$ (CD8 stimulated) individual experiments. Data depicted as mean $\pm$ SEM (OCR) and box plots with median, 25$^{th}$ and 75$^{th}$ percentiles and range (all other). Dots indicating individual values. *$P < 0.05$, **$P < 0.01$, paired t-test/Wilcoxon matched-pairs signed rank test, as appropriate.

of mitochondrial superoxide were found (Figs 3B and EV2E). These changes were limited to activated T cells (Fig EV2A and B).

Increased mROS may be the result of amplified oxidative phosphorylation. Indeed, we observed increased mitochondrial mass in BHB$^+$ CD4$^+$ and CD8$^+$ T cells (Figs 3C and D, and EV2F), and Western blot analysis revealed enhanced expression of ETC complexes in BHB$^+$ primary human T cells and CD4$^+$/CD8$^+$ T-cell subsets (Figs 3E and EV2G). With respect to hormesis, elevated (mitochondrial) ROS have also been associated with mitochondrial and cellular damage (Zorov et al, 2014). However, an increase in GSH levels shows that protective antioxidative capacity also improved in BHB$^+$ CD4$^+$ and CD8$^+$ T cells (Fig 3F). To further exclude structural impairment of mitochondria evoked by ketogenic conditions, we assessed the mitochondrial integrity of these cells. As depicted in Fig 3G, BHB did not alter mitochondrial membrane potential of primary human T cells, neither in CD4$^+$ nor in CD8$^+$ T-cell subsets. Hence, ketone metabolism does not compromise mitochondrial integrity.

Collectively, we have shown an increase in mitochondrial mass, ROS production, and aerobic oxidative metabolism through enhanced respiratory chain activity in BHB$^+$ human T cells. This might result in an augmentation of memory T cell (T$_{mem}$) formation as they primarily rely on OXPHOS, and increased spare respiratory capacity has been viewed as a hallmark of their immunometabolism (van der Windt et al, 2012; O'Sullivan et al, 2014). Supporting this hypothesis, BHB-cultivated cells expressed elevated levels of T$_{mem}$ differentiation factor IL15 (Fig 3H). We applied a T$_{mem}$ differentiation protocol to further investigate the impact of ketone bodies on T$_{mem}$ development (Fig 3I). As shown in Fig 3J, BHB$^+$ CD4$^+$ and CD8$^+$ T cells displayed higher expression of IL7R and IL15 after differentiation. CD4$^+$ and CD8$^+$ central and effector memory T-cell (T$_{cm}$/T$_{em}$) subset analysis revealed a significant increase in CCR7$^-$CD45RA$^-$CD45RO$^+$CD8$^+$ T$_{em}$ (Fig 3K). In summary, an enhanced aerobic mitochondrial metabolism in response to BHB directs human T cells toward T$_{mem}$ differentiation.

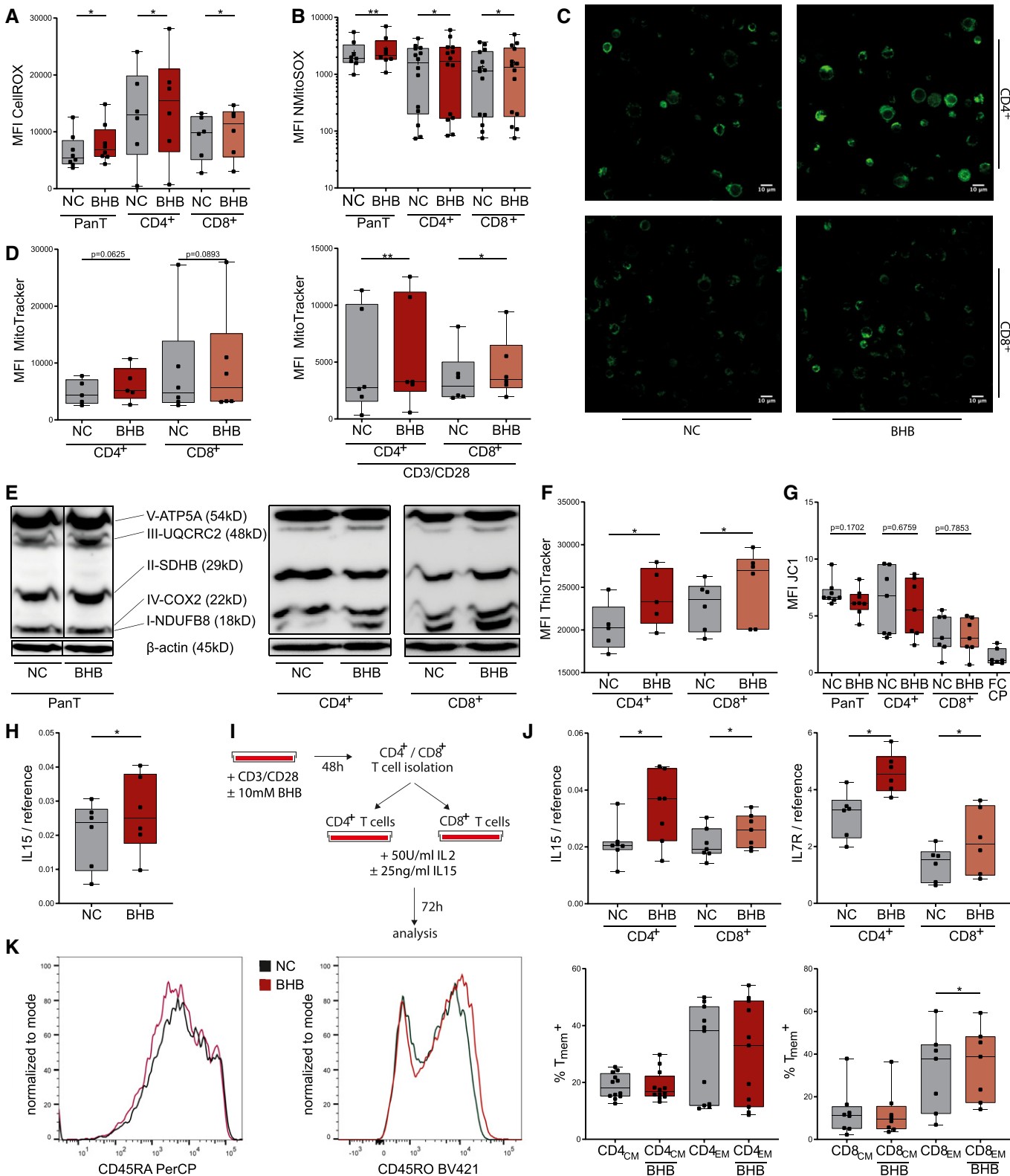

**Figure 3.**

◀

**Figure 3. Beta-hydroxybutyrate increases ROS formation in human T cells and directs them to memory cell formation *in vitro*.**

Human peripheral blood mononuclear cells (PBMCs) were cultivated for 48 h in RPMI containing 80 mg/dl glucose (NC) and supplemented with 10 mM beta-hydroxybutyrate (BHB). T-cell stimulation was performed through CD3/CD28 Dynabeads at a bead:cell ratio of 1:8. Flow cytometry analyses were performed using the indicated dyes and antibodies identifying T cells and $CD4^+/CD8^+$ T-cell subsets. For JC1 analysis, magnetic $CD4^+/CD8^+$ T-cell separation was performed prior to staining.

A Quantification of cellular reactive oxygen species (ROS) using CellROX dye, indicated by mean fluorescence intensity (MFI) FITC in human pan/$CD4^+/CD8^+$ T cells, $n = 8/6/6$ biological replicates.

B Mitochondrial superoxide production quantified through MitoSOX staining, depicted as MFI phycoerythrin (PE) in human pan/$CD4^+/CD8^+$ T cells, $n = 8/14/14$ biological replicates.

C Confocal microscopy images of human $CD4^+/CD8^+$ T cells, stained with MitoTracker green, representative of two individual experiments.

D Quantification of mitochondrial mass using MitoTracker green, indicated by MFI FITC in human $CD4^+/CD8^+$ T cells +/− CD3/CD28 stimulation, $n = 5/6/6/6$ biological replicates.

E Western blot of mitochondrial oxidative phosphorylation proteins in pan/$CD4^+/CD8^+$ T cells as indicated, representative of at least three individual experiments.

F Quantification of cellular glutathione content using ThioTracker, indicated by MFI FITC in human $CD4^+/CD8^+$ T cells, $n = 5/6$ biological replicates.

G Evaluation of mitochondrial membrane potential of human pan/$CD4^+/CD8^+$ T cells using quantification of JC1 fluorescence (MFI PE/FITC), FCCP served as the negative control, $n = 8/7/7/7$(FCCP) biological replicates.

H IL15 mRNA expression in stimulated pan T cells, $n = 6$ biological replicates.

I T memory cell differentiation protocol: Following initial incubation and stimulation, $CD4^+/CD8^+$ T cells were isolated via magnetic cell separation and cultivated with 50 U/ml IL2 and 25 ng/ml IL15 for an additional 72 h.

J IL15 and IL7R mRNA expression in $CD4^+/CD8^+$ T cells after memory T-cell differentiation, $n = 7/6$ biological replicates.

K Flow cytometric quantification of memory T cells: $CD4^+/CD8^+$ T cells were stained for CCR7 ($PE^{+/−}$) and subsequently defined memory phenotype for $CD45RO^+$ (Pacific $Blue^+$) and $CD45RA^-$ ($PerCP^-$) staining. Histogram plots depicting the exemplary change of CD45RO and CD45RA distribution for $CD8^+$ T cells following memory cell differentiation (left side; NC = black, BHB = red). Fractions of $CCR7^+CD45RA^-CD45RO^+$ central memory (CM) and $CCR7^-CD45RA^-CD45RO^+$ effector memory (EM) $CD4/8^+$ T cells (right side), $n = 12/11/8/7$ biological replicates.

Data information: Data depicted as box plots with median, 25th and 75th percentiles and range. Dots indicating individual values. $*P < 0.05$, $**P < 0.01$, paired *t*-test/ Wilcoxon matched-pairs signed rank test, as appropriate.

Source data are available online for this figure.

## Ketogenic diet *in vivo*

The impact of KD on human immune responses *in vivo* has not yet been investigated. We conducted a prospective immuno-nutritional *in vivo* study enrolling 44 healthy volunteers, who performed a 3-week KD. Blood was collected prior to the start ($T_0$) and at the end ($T_1$) of KD. Blood ketone body levels were closely monitored. The study design is depicted in Fig 4A; characteristics of study participants are given in Table 1. During the course of KD, a significant increase in blood ketone bodies (in the range of 1.0–1.5 mM) was detected until the 2nd week, which could be sustained until the end of the study (Fig EV3A). Fasting serum glucose, however, remained unchanged throughout the course of a 3-week diet (Fig EV3B). Apart from a brief phase of increased fatigue (duration 1–5 days), no relevant side effects were reported (Table 1). KD led to a significant decrease in participants' body mass index (BMI; Fig EV3C). However, this effect was mainly obtained by overweight volunteers, who achieved a profound weight loss, while participants with $T_0$ BMI < 25 experienced only marginal BMI changes (Table 1).

## Transcriptome and gene set enrichment analysis reveals immunometabolic reprogramming of human T cells under the influence of KD

*In vitro* results indicated substantial alterations to T-cell immunometabolism in response to BHB. To analyze the global impact of KD on gene expression *in vivo*, transcriptome profiles of human T-cell RNA samples from healthy volunteers conducting a 3-week KD were generated (Fig 4B). Differential expression analysis $T_1/T_0$ revealed a significant regulation of gene expression in both $CD4^+$ and $CD8^+$ T cells in response to KD (Fig 4C and D). $T_1$ T cell transcriptome reflected enhanced immune competence with upregulation of both genes related to T-cell activation and effector function (TNF, TNFRSF9, CRTAM, DUSP4, TRAV30) but also inhibitory and regulatory capacity (EGR2, NFKBID). Importantly, enrichment of TCR-related genes TRAV30 and NR4A1 (Nur77) was also detected. Nur77 is known to indicate T-cell receptor signaling strength and is associated with memory cell development (Li *et al*, 2020; Shin *et al*, 2020). Notably, Nur77 has recently been identified as a central regulator of T-cell immunometabolism (Liebmann *et al*, 2018). Furthermore, genes involved in mitochondrial function were also differentially expressed on a KD (NR4A1, TOMM7, ACSL6).

These findings were substantiated by gene set enrichment analysis (GSEA; Fig 4E). Several immune response pathways—involving both inflammatory and regulatory gene sets—were upregulated in response to KD. Major alterations could also be detected for metabolic gene sets. In summary, these results highlight substantial transcriptomic changes in favor of T-cell action, memory cell differentiation, and oxidative metabolism, thus pointing to immunometabolic reprogramming of human $CD4^+$ and $CD8^+$ T cells through KD.

## KD enhances human T-cell immune capacity

We subsequently investigated the immunometabolic phenotype of human T cells in response to KD (KD T cells), which revealed similar changes as found *in vitro*. In unstimulated $CD4^+/CD8^+$ T cells, only minimal transcriptional activity and no relevant change of T-cell cytokine levels $T_0/T_1$ was detected (Appendix Fig S2A and B), whereas *ex vivo* activated $T_1$ $CD4^+$ T cells displayed an upregulation of IL2, IL4, and CTLA4 mRNA (Fig 5A). KD led to an elevation of $Th_2$ transcription factor GATA3 (Fig 5B) resulting in a decline of Tbet/GATA3 ratio (Appendix Fig S2B), an augmentation of $Th_2$ T helper cell subset and a decline in $Th_1/Th_2$ cell ratio (Fig 5C). $T_1$

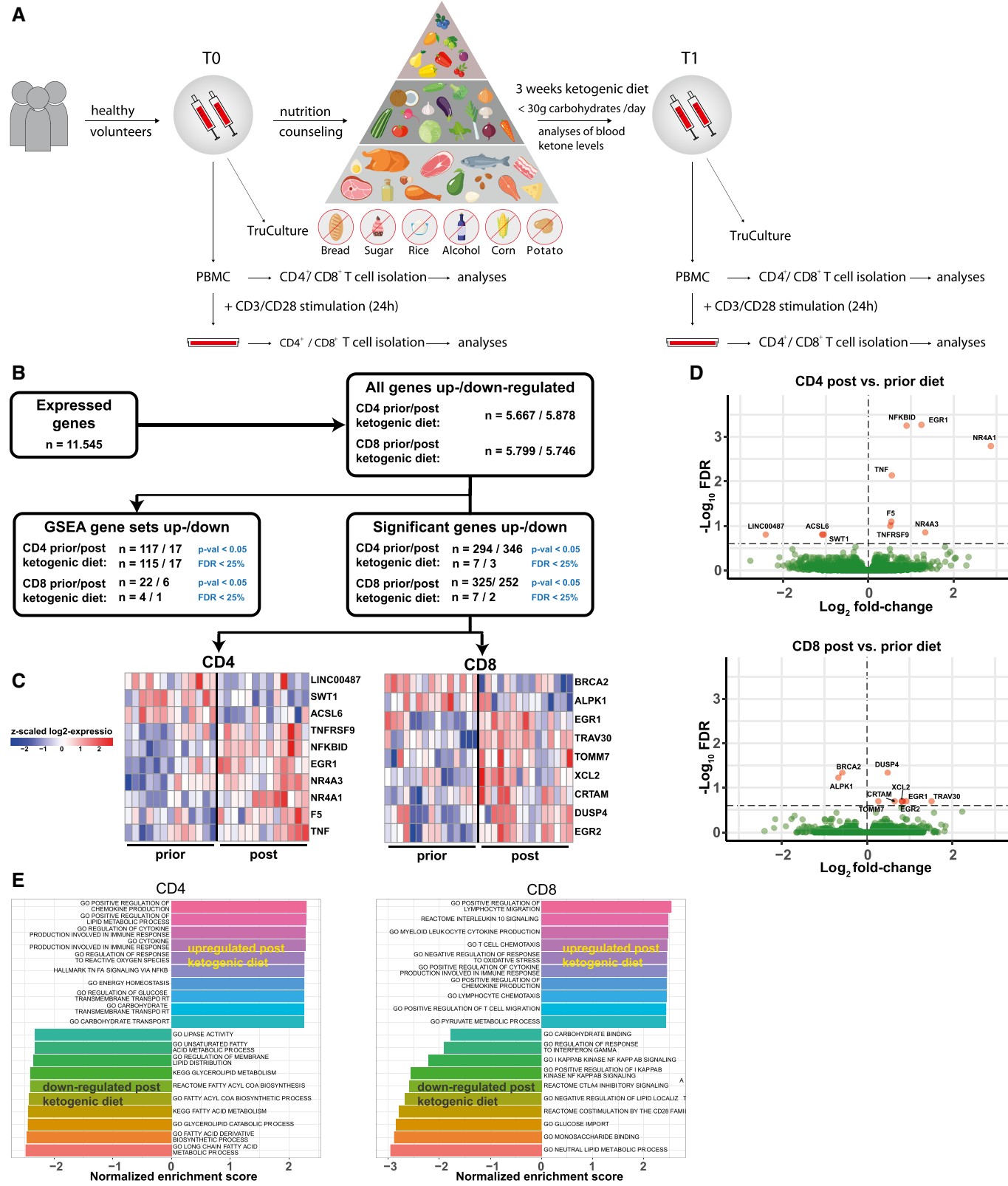

Figure 4.

**Figure 4. Next-generation sequencing reveals immunometabolic reprogramming in human T cells on KD.**

A   Schematic study flow: 44 healthy volunteers conducted a 3-week ketogenic diet (KD) with limited carbohydrate consumption of < 30 g/day. Blood was taken and analyzed before starting the diet ($T_0$) and after 3 weeks of successful diet ($T_1$). PBMCs were isolated. T-cell stimulation was performed through CD3/CD8 Dynabeads at a bead:cell ratio of 1:8. CD4$^+$/CD8$^+$ T cells were separated via magnetic cell labeling.

B   Total data set gene expression and differential regulation at the gene and pathway levels in human CD4$^+$/CD8$^+$ T cells prior to and post-KD. Differential expression analysis was performed using DESeq2 while statistical significance was accepted for corrected $P$-values (FDR) smaller than 25%.

C   Heatmaps of $z$-scaled expressions of significantly differentially expressed genes in CD4$^+$ T cells (left) and CD8$^+$ T cells (right) prior to and post-KD. Red color indicates an upregulation post-KD, and downregulated genes are indicated by blue color.

D   Volcano plots visualizing differential gene expression $T_1/T_0$ in human CD4$^+$/CD8$^+$ T cells. $\log_2$ fold-changes ($x$-axis) and $-\log_{10}$ FDR (false discovery rate) corrected $P$-values ($y$-axis) are shown for each gene. The dashed horizontal line depicts the FDR threshold of 25%, significant genes in red.

E   Gene set enrichment analysis. Shown are the top 10 upregulated and downregulated pathways ranked according to normalized enrichment score (NES).

**Table 1. Characteristics of healthy volunteers. All healthy volunteers were without suspicion of acute or chronic disease. Blood count and electrolytes were within normal range.**

| $n$ | 44 |
|---|---|
| Age, years (mean ± SD) | 35.56 (± 11.56) |
| Sex (female/male/divers) | f = 25 \| m = 19 |
| Blood BHB, day 07, mM (mean ± SEM) | 0.998 (± 0.092) |
| Blood BHB, day 14, mM (mean ± SEM) | 1.34 (± 0.104) |
| Blood BHB, day 21, mM (mean ± SEM) | 1.40 (± 0.158) |
| Fasting Blood Glucose T0, mg/dl (mean ± SEM) | 93.67 (± 4.04) |
| Fasting Blood Glucose T1, mg/dl (mean ± SEM) | 98.61 (± 4.65) |
| BMI T0, kg/m$^2$ (mean ± SEM) | 23.77 (± 0.524) |
| BMI T1, kg/m$^2$ (mean ± SEM) | 22.83 (± 0.437) |
| Overweight volunteers | |
| BMI T0, kg/m$^2$ (mean ± SEM) | 27.65 (± 0.701) |
| BMI T1, kg/m$^2$ (mean ± SEM) | 25.59 (± 1.022) |
| Normal weight volunteers: | |
| BMI T0, kg/m$^2$ (mean ± SEM) | 22.59 (± 0.401) |
| BMI T1, kg/m$^2$ (mean ± SEM) | 21.99 (± 0.324) |
| Side effects (0 = no occurrence \| 1 = rarely \| 2 = occasionally \| 3 = often): | |
| Nausea | 0: 91% \| 1: 9% |
| Vomiting | 0: 95% \| 1: 5% |
| Constipation | 0: 86% \| 1: 14% |
| Diarrhea | 0: 89% \| 1: 11% |
| Early fatigue (day 1–5) | 0: 25% \| 1: 5% \| 2: 16% \| 3: 50% |
| Persistent fatigue (day 6–21) | 0: 95% \| 1: 5% |
| Reduced physical performance | 0: 91% \| 1: 5% \| 2: 5% |
| Headache | 0: 86% \| 1: 14% |
| Vertigo | 0: 100% |

$T_{reg}$ subpopulation was also enhanced as shown by a higher abundance of CD4$^+$CD25$^+$Foxp3$^+$ T cells and an upregulation of IL10 mRNA (Fig 5D). Moreover, $T_1$ CD8$^+$ T cells exhibited elevated IFNγ, GZMB, and CTLA4 cytokine levels and significantly increased cell lysis activity (Fig 5E and F). Thus, these data demonstrate an enhanced human T-cell immune capacity through KD *in vivo*.

## KD strengthens mitochondrial metabolism and memory cell development

We next analyzed mitochondrial respiratory chain activity in human KD T cells. After 3 weeks of KD, $T_1$ CD4$^+$ T cells exhibited upregulated basal oxidative respiration (Fig 6A). For $T_1$ CD8$^+$ T cells, profound changes were detected: elevated basal and maximal respiratory rates as well as augmented mitochondrial ATP production and spare respiratory capacity (Figs 6B and EV4A). Of note, this superior oxidative capacity was not at the expense of glycolytic capabilities as seen for extracellular acidification and glycolytic proton efflux (Fig EV4B–G). These metabolic alterations—leading to emphasized oxidative phosphorylation—translated into elevated cellular and mitochondrial ROS in $T_1$ T cells (Fig 6C). Also *in vivo*, no impairment of mitochondrial membrane integrity occurred (Fig EV4H). Cellular increase in ROS was restricted to T-cell activation, whereas higher mROS could also be detected in unstimulated $T_1$ T cells (Fig EV4I and J). Consequently, $T_1$ CD4$^+$ and $T_1$ CD8$^+$ T cells displayed higher mitochondrial mass and augmented expression of OXPHOS protein complexes (Figs 6D–F and EV4K). We then evaluated the effects of KD on memory cell development in human T cells *in vivo*. Memory cell subset analysis revealed elevated levels of both CD4$^+$ and CD8$^+$ CCR7$^-$CD45RA$^-$CD45RO$^+$ effector memory cells ($T_{Em}$; Fig 6G). In summary, KD led to an increase in mitochondrial mass, ROS production, and aerobic oxidative metabolism through enhanced respiratory chain activity. These KD-induced changes in human T-cell immunometabolism promote $T_{mem}$ development.

## KD improves overall immune responses *in vivo*

Immune responses result from a complex interplay of innate and adaptive mechanisms. To explore the full potential of KD *in vivo*, we analyzed TruCulture whole blood protein levels, which allow the detection of overall immune responses of adaptive and innate immune cells in a physiological environment during pathogen encounter. KD led to a significant upregulation of IFNγ, IL4, IL6, IL12sub40, IL23, and TNFα (Fig 6H) and thus confirmed the enhancement of human adaptive immune response by KD. Importantly, innate immune master cytokines IL1α/β displayed unaltered expression levels (Fig EV4L). TruCulture analysis of unstimulated whole blood samples also ruled out immunostimulatory effects of KD *per se* (Fig EV4M).

In conclusion, KD significantly reshaped human T-cell immunity toward a more powerful yet controlled adaptive immune response.

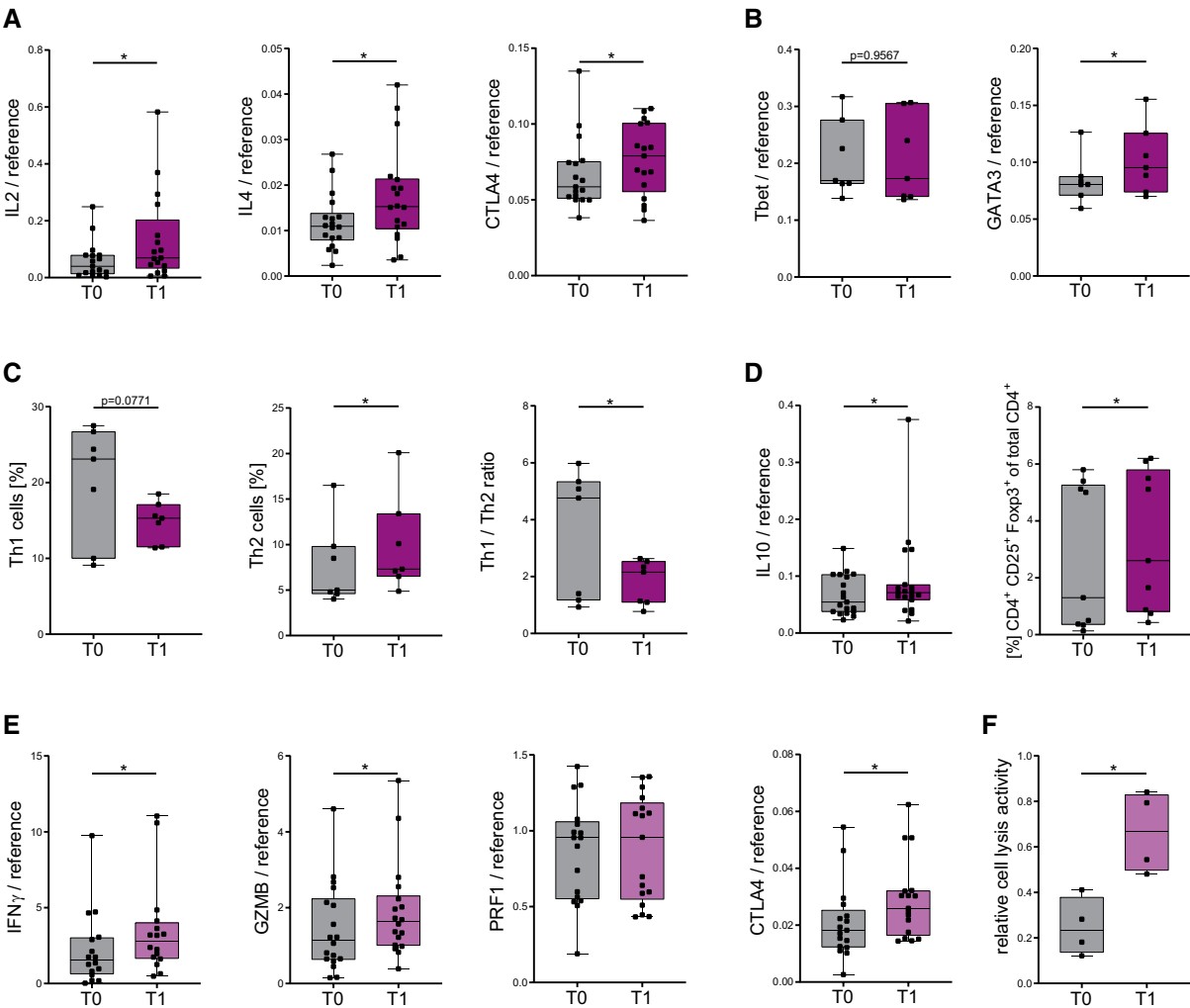

**Figure 5. Ketogenic diet enhances human T-cell immune capacity *in vivo*.**

Healthy volunteers conducted a 3-week KD with a limited carbohydrate consumption of < 30 g/day. Blood was taken and analyzed prior to (T$_0$) and post-KD (T$_1$). PBMCs were isolated. T-cell stimulation was performed through CD3/CD28 Dynabeads at a bead:cell ratio of 1:8. CD4$^+$/CD8$^+$ T cells were separated via magnetic cell labeling.

A   IL2, IL4, and CTLA4 mRNA expression in stimulated CD4$^+$ T cells, *n* = 17/18/17 individual human subjects.
B   Tbet and GATA3 mRNA expression in CD4$^+$ T cells, *n* = 7 individual human subjects.
C   Flow cytometric quantification of Th$_1$/Th$_2$ cells and respective ratio of Th$_1$/Th$_2$ cells, *n* = 7 individual human subjects.
D   IL10 mRNA expression and flow cytometric quantification of CD4$^+$CD25$^+$Foxp3$^+$ regulatory T cells (T$_{reg}$), *n* = 19/9 individual human subjects.
E   IFNγ GZMB, PRF1, and CTLA4 mRNA expression in CD8$^+$ T cells, *n* = 16/18/17/17 individual human subjects.
F   Relative CD8$^+$ cell lysis activity as measured by calcein fluorescence of isolated T cells, *n* = 4 individual experiments.

Data information: Data depicted as box plots with median, 25$^{th}$ and 75$^{th}$ percentiles and range. Dots indicating individual values. *$P < 0.05$, paired *t*-test/Wilcoxon matched-pairs signed rank test, as appropriate.

# Discussion

Western diet is increasingly recognized as a true endangerment to public health. It accounts for a rapid increase in obesity, diabetes, cardio- and neurovascular diseases, and even cancer (Mattson *et al*, 2014; Cohen *et al*, 2015; Ludwig, 2016; Hotamisligil, 2017). Chronic low-grade inflammation induced by both adipose tissue and forced dietary uptake of carbohydrates is considered a main driver of these conditions. The resulting unspecific activation of the innate immune system is not only harmful in itself, but also strongly impairs adaptive immune responses and hampers the ability to create

immunological memory (Neidich *et al*, 2017; Napier *et al*, 2019; Ritter *et al*, 2020).

Modern medicine has to establish new strategies to enable immunometabolic reprogramming to prevent and even treat these damaging conditions. At this point, nutritional interventions as a clinical tool enter the stage. One current and highly discussed approach is KD, a high-fat low-carbohydrate KD. Ketosis represents an evolutionary conserved physiological state characterized by reduced carbohydrate uptake leading to moderate hepatic production of ketone bodies ("ketosis", 0.5–5 mM BHB) (Krebs, 1966). KD is considered a tool to control body weight and has been shown to

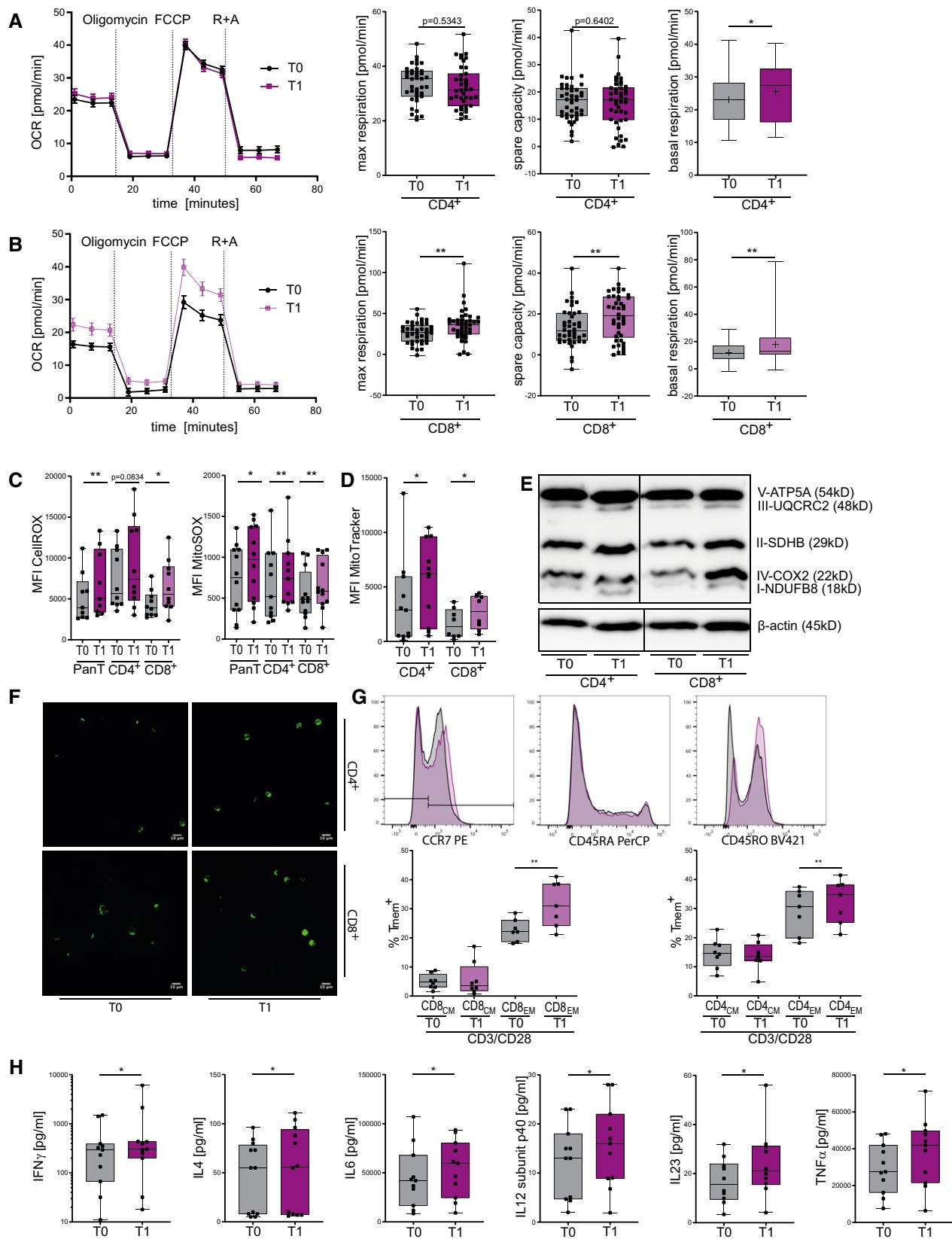

**Figure 6.**

**Figure 6.  Ketogenic diet primes human T cells to mitochondrial metabolism and memory cell development *in vivo*.**

Forty-four healthy volunteers conducted a 3-week KD with a limited carbohydrate consumption of < 30 g/day. Blood was taken and analyzed prior to starting the diet ($T_0$) and again after 3 weeks of strict adherence to the diet ($T_1$). PBMCs were isolated. T-cell stimulation was performed through CD3/CD28 Dynabeads at a bead:cell ratio of 1:8. Pan/CD4$^+$/CD8$^+$ T cells were separated via magnetic cell labeling. Mitochondrial metabolism was analyzed for each subpopulation using a Seahorse XF96 Analyzer.

A, B   OCR, maximum respiration, spare respiratory capacity, and basal respiration were measured in (A) CD4$^+$ and (B) CD8$^+$ T cells, $n$ = 14 individual experiments, each performed in three technical replicates.

C   Quantification of cellular and mitochondrial ROS using CellROX/MitoSOX, indicated by MFI FITC/MFI PE in human Pan/CD4$^+$/CD8$^+$ T cells, $n$ = 9/10/10 (CellROX), 12/11/12 (MitoSOX) individual human subjects.

D   Quantification of mitochondrial mass using MitoTracker green, indicated by MFI FITC in human CD4$^+$/CD8$^+$ T cells, $n$ = 11/8 individual human subjects.

E   Western blot of mitochondrial oxidative phosphorylation proteins in CD4$^+$/CD8$^+$ T cells as indicated, representative of five individual experiments.

F   Confocal microscopy images of human CD4$^+$/CD8$^+$ T cells, stained with MitoTracker green, representative of two individual experiments.

G   Flow cytometric quantification of memory T cells *in vivo*: CD4$^+$/CD8$^+$ T cells were stained for CCR7 (PE$^{+/-}$) and subsequently defined memory phenotype for CD45RA$^-$ (PerCP$^-$) and CD45RO$^+$ (Pacific Blue$^+$) staining, representative histogram plots (top), T0 = gray, T1 = purple. Fractions of CCR7$^+$CD45RA$^-$CD45RO$^+$ central memory (CM) and CCR7$^-$CD45RA$^-$CD45RO$^+$ effector memory (EM) CD4$^+$/8$^+$ T cells (bottom), $n$ = 8/7 (CM/EM) individual human subjects.

H   TruCulture IFNγ, IL4, IL6, IL8, IL12 subunit p40, IL23, and TNFα protein quantification of LPS-stimulated whole blood samples, $n$ = 11/12/11/10/11/10/11 individual human subjects.

Data information: Data depicted as mean ± SEM (OCR) or as box plots with median, 25$^{th}$ and 75$^{th}$ percentiles and range (all others), crosses indicating mean (A and B). Dots indicating individual values. $*P < 0.05$, $**P < 0.01$, paired *t*-test/Wilcoxon matched-pairs signed rank test, as appropriate.

Source data are available online for this figure.

potently ameliorate inflammatory processes in mouse models (Paoli *et al*, 2013; Youm *et al*, 2015; Goldberg *et al*, 2017). To date, however, the impact of KD on human immunity remains elusive.

We present the first study investigating the effects of KD on human immune cells *in vitro* and in a prospective series of healthy subjects. We did not use murine models due to their substantial inherent limitations with respect to accurately reflecting human immune responses (Pulendran & Davis, 2020). We report profound immunometabolic effects of KD on human lymphocytes both *in vitro* and *in vivo*. KD T cells displayed enhanced reactivity upon specific stimulation, strengthened cell lysis capacity, increased differentiation of regulatory T cells, and pronounced T memory cell formation. KD promoted mitochondrial mass and expression of ETC complexes, enabling augmented mitochondrial oxidative phosphorylation leading to moderately increased cellular and mitochondrial ROS production and superior respiratory reserve. These alterations were accompanied by global transcriptomic changes of human CD4$^+$ and CD8$^+$ T cells, linking KD to a fundamental immunometabolic reprogramming. In contrast to previous mouse studies, innate immune responses were not compromised by KD, which further highlights the need for human *in vivo* data.

For the *in vivo* part of our study, healthy participants performed an individual 3-week *ad libitum* KD, resulting in blood ketone levels in the range of 1.0–1.5 mM. No relevant adverse effects were reported. Of note, morning fasting glucose levels remained in normal range during the course of the 3-week diet, most likely due to hepatic gluconeogenesis, consistent with previous data (Cahill & Veech, 2003). It can be assumed that longer periods of KDs are needed to achieve a stable lowering of blood glucose levels. BMI of overweight study participants (BMI>25) was significantly reduced after the study period, while the body weight of non-overweight subjects remained almost unaffected. This could indicate a particular responsivity of adipose tissue to the reduction of blood glucose peaks and subsequent reduced insulin secretion. However, in our series, covering a time span of 3 weeks, no significant differences in the immunological outcome between overweight and non-overweight participants were detectable, which underlines our *in vitro* findings pointing out direct immunomodulatory effects of BHB. Longer KD periods in the specific setting of obesity are

probably required to detect potential indirect effects resulting from the reduction of adipose tissue.

T cells have been shown to incorporate BHB-derived acetyl-CoA into the TCA cycle (Zhang *et al*, 2020). Compared to fatty acid oxidation (FAO), metabolization of BHB displays higher efficiency due to NADH-based delivery of all reducing equivalents to complex I of the ETC (Cotter *et al*, 2013). Elevated energy supply through metabolization of BHB may thus improve T-cell immunity. We, indeed, found that use of BHB as an alternative energy fuel enhanced mitochondrial metabolism and increased cellular power reserves. Activated T cells obtain a large proportion of their ATP production through aerobic glycolysis both *in vitro* and *in vivo* (Michalek *et al*, 2011; Chang *et al*, 2013; Macintyre *et al*, 2014; Buck *et al*, 2016). Importantly, the here reported upregulation of mitochondrial oxidative phosphorylation was not on the cost of glycolysis but on top, as we observed unchanged glycolytic capacity of human KD T cells. As a result, higher overall levels of biochemical energy are yielded. It is conceivable that KD could also increase glutaminolysis, another ATP-generating pathway used by T cells (Wang *et al*, 2011; Wang & Green, 2012), for example, by increasing activity of glutamate dehydrogenase via SIRT4 inhibition (Li *et al*, 2011; Min *et al*, 2018). As T cells—in contrast to tumors—are capable of glutamine oxidation, it can be assumed that ATP production as measured by Seahorse analysis yields reliable results. However, the contribution of glutamine fermentation to cellular ATP production escapes detection via Seahorse analysis and may thus further contribute to T-cell energy content. Detailed deciphering of ATP-delivering pathways affected by KD requires further investigation.

Effector T cells may additionally be supported by elevated mitochondrial reactive oxygen species, which are required to mount an antigen-specific immune response (Sena *et al*, 2013). The role of ROS in T-cell immunity is exceptional and markedly differs from tissues and particularly from cancer cells: ETC-derived ROS serve as a second messenger during T-cell activation (Murphy & Siegel, 2013; Franchina *et al*, 2018), thus being considered pivotal for T-cell immunity (Devadas *et al*, 2002; Jones *et al*, 2007; Sena *et al*, 2013). Of note, T cells diverge from NADPH-dependent GSH synthesis, instead redirecting NADPH for ROS production via NADPH oxidase

to fulfill their demand on reactive oxygen species (Jackson et al, 2004; Kwon et al, 2010). These findings gave rise to the concept of mitohormesis, opposing the idea of ROS as solely detrimental byproducts of an imperfect oxidative system, but emphasizing the role of ROS as essential signaling molecules (Ristow, 2014). In non-phagocytic cells, ETC complexes are known to account for the majority of ROS production (Desdín-Micó et al, 2018; Raimondi et al, 2020). We show that KD leads to an amplified expression and activity of ETC complexes in human T cells, creating higher levels of mitochondrial and—in case of T-cell activation—cellular ROS. A mild increase in ROS production has already been associated with protective effects of KD via adaptational upregulation of antioxidative capabilities and amelioration of oxidative stress, which is in line with the observed augmentation of GSH levels and the preservation of mitochondrial membrane potential during KD (Milder & Patel, 2012). Consequently, both enhanced overall energy supply and (m) ROS-signaling create the bioenergetic basis for augmented T-cell immune responses in KD T cells.

Importantly, KD revealed also as a possible strategy to prime T cells toward T memory cell formation. We discovered an increase in mitochondrial mass, spare respiratory capacity (SRC)—both considered hallmarks of $T_{mem}$ cells—and, indeed, an enrichment of T memory cell formation in vivo and in vitro (van der Windt et al, 2012, 2013). It has previously been reported that T memory cell development depends on oxidative metabolism to ensure superior respiratory reserve, sustaining higher levels of ATP upon reactivation (Buck et al, 2016; Simula et al, 2017). Moreover, T cells are known to perform active ketogenesis to utilize BHB for epigenetic modifications to support memory cell development (Zhang et al, 2020).

Ketogenic diet could also prove to be a valuable measure in the threatening situation of the current COVID-19 pandemic, as dysregulation and exhaustion of both CD4$^+$ and CD8$^+$ T cells as well as impairment of classical CD8$^+$ T effector memory cells have been reported as central elements of COVID-19 immunopathology (Chen & John Wherry, 2020; Habel et al, 2020). Thus, clinical evaluation could be promising.

Our study demonstrates that KD induces a fundamental immunometabolic reprogramming in human T cells associated with profound transcriptomic changes. This leads to a balanced global enhancement of T-cell immunity, comprising enhanced cytokine production and secretion, strengthened cell lysis capacity, amplified Treg differentiation, and pronounced Tmem cell formation. KD thus holds promise as a feasible and effective clinical tool for a large range of conditions intimately associated with immune disorders. This provides the basis to proceed into further medical translation. Consequently, a clinical phase II study investigating KD in sepsis patients is currently recruiting (Rahmel et al, 2020).

These new immunological aspects of KD might contribute to the modern concept of metabolic therapy of cancer (Seyfried et al, 2017, 2020). KD not only targets the Warburg effect, but could also strengthen anti-tumor immunity (Ferrere et al, 2021). Moreover, the different effects of BHB-induced ROS elevation on tumor cells and T cells might even lead to additive beneficial effects: While oxidative stress compromises cancer cell viability, mild increase in ROS enhances T-cell immune capacity which in turn further restrains tumor growth. However, these issues need to be addressed in future clinical studies.

In conclusion, our study changes the perspective on nutrition as a clinical tool and could help to redefine the role of dietary interventions in modern medicine.

# Material and Methods

### In vitro peripheral blood mononuclear cell culture and stimulation

Peripheral blood mononuclear cells (PBMCs) from healthy volunteers not yet undertaking the specified KD were obtained via density centrifugation (Histopaque-1077; Sigma-Aldrich, St. Louis, MO, USA). To assess cell numbers and viability, a ViCell analyzer (Beckman Coulter, Fullerton, CA, USA) was utilized. PBMCs were cultivated in RPMI 1640 (Invitrogen, Carlsbad, CA, USA) containing 80 mg/dl glucose supplemented with 10% heat-inactivated fetal calf serum (Biochrom, Berlin, Germany), 1% HEPES (Sigma-Aldrich, St. Louis, MO), 1% L-glutamine (Life Technologies, Carlsbad, CA, USA), 100 U/ml penicillin, and 100 U/ml streptomycin. BHB-containing medium for PBMC cultivation was prepared by adding 10 mM beta-hydroxybutyrate (BHB; Sigma-Aldrich, St. Louis, MO, USA) to the modified RPMI 1640 specified above. Cells were incubated at 37°C and 5% $CO_2$. T-cell fractions were stimulated using 50 U/ml IL2 and CD3/CD28 Dynabeads (Thermo Fisher Scientific, Waltham, MA, USA) with a bead-to-cell ratio of 1:8 for a duration of 48 h.

### In vitro whole blood cell culture and stimulation

Whole blood samples were obtained from fasting healthy donors and incubated at 37°C. BHB cultivation was performed by adding 10 mM beta-hydroxybutyrate (BHB; Sigma-Aldrich, St. Louis, MO, USA) to the whole blood samples. Immune cells were stimulated using 100 ng/ml Lipopolysaccharide (LPS) for a duration of 3 h. Prior to subsequent analyses, red blood cell lysis was performed using Red Blood Cell Lysis Buffer (Invitrogen, Carlsbad, CA, USA) according to the manufacturer's instructions.

### In vivo study design

Healthy volunteers participated in a prospective study investigating the effects of a KD. All participants were nonsmokers and not suspected of suffering from any acute or chronic diseases. Prior to starting the diet, all volunteers attended nutritional counseling. Study participants adhered to a KD consuming less than 30 g carbohydrates per day for a duration of 3 weeks. Ketone body levels were closely monitored by point of care testing using a Glucomen Aero 2K (Berlin Chemie AG, Berlin, Germany). Serum glucose levels were measured via the hexokinase method using a Cobas 8000/c702 (Roche Diagnostics, Penzberg, Germany). Prior to the start ($T_0$) and on the last day of the KD ($T_1$), blood samples were collected from all volunteers. Informed consent was obtained from all volunteers. Research was performed according to the Declaration of Helsinki (ethical principles for medical research involving human subjects) and the U.S. Department of Health and Human Services Belmont Report. The study design and the study protocol were approved by the Institutional Ethics Committee of the Ludwig-Maximilian-

University Munich, Germany (No. 19-523). The study was registered at the DKRS (German Clinical Trials Register; DRKS-ID: DRKS00023373). Power analysis was performed based on *in vitro* results. Interferon γ expression $T_1/T_0$ was designated as the primary study endpoint. A sample size of $n = 40$ was calculated presuming an estimated effect size (Cohen's *d*) of 0.5, a level of significance α = 0.05, a drop-out rate of 20%, and a power level of 80%. Characteristics of healthy volunteers are depicted in Table 1.

### *Ex vivo* peripheral blood mononuclear cell stimulation and T-cell isolation

Peripheral blood mononuclear cells were obtained by density centrifugation (Histopaque-1077; Sigma-Aldrich, St. Louis, MO, USA). A ViCell analyzer (Beckman Coulter, Fullerton, CA, USA) was used to evaluate the cell count and viability. PBMCs were subjected to immediate T-cell isolation as well as cell cultivation in RPMI 1640 (Invitrogen, Carlsbad, CA, USA) containing 10% heat-inactivated fetal calf serum (Biochrom, Berlin, Germany), 1% HEPES (Sigma-Aldrich, St. Louis, MO), and 1% L-glutamine (Life Technologies, Carlsbad, CA, USA). T cells were stimulated via the addition of CD3/CD28 Dynabeads (Thermo Fisher Scientific, Waltham, MA, USA) with a bead-to-cell ratio of 1:8 for a duration 24 h.

### T-cell isolation

After stimulation, CD3/CD28 Dynabeads were magnetically removed. Pan T-cell, CD4[+]-cell, and CD8[+]-cell isolation was performed by magnetic cell separation (Pan T Cell Isolation Kit, # 130-096-535 | human CD4 MicroBeads, # 130-045-101 | human CD8 MicroBeads, # 130-045-201, Miltenyi Biotec, Bergisch Gladbach, Germany) using an AutoMACS Pro Separator (# 130-092-545, Miltenyi Biotec, Bergisch Gladbach, Germany) according to the manufacturer's instructions.

### $T_{reg}$ differentiation

For regulatory T-cell ($T_{reg}$) differentiation, CD4[+] T cells were incubated for 5 days using TexMACS[TM] medium (# 130-097-196, Miltenyi Biotec, Bergisch Gladbach, Germany), stimulated with T Cell TransAct[TM] human (Miltenyi Biotec, Bergisch Gladbach, Germany) and supplemented with 5 ng/ml TGFβ1, 1 µg/ml anti-IFNγ, 1 µg/ml anti-IL4, 100 U/ml IL2 (#130 095-067 | #130 095-743 | # 130-095-709 | #130 097 742, Miltenyi Biotec, Bergisch Gladbach, Germany), and 10 mM of retinoic acid ( #R2625; Sigma-Aldrich, Darmstadt, Germany).

### T helper cell subset differentiation

For differentiation into T helper cell 1/2 subsets, CD4[+] T cells were incubated for 96 h in cell culture plates covered with anti-human CD3 (#300437, BioLegend, San Diego, CA) using TexMACS[TM] medium. Cells were stimulated with T Cell TransAct[TM] human as described above. Specific cytokines were added for Th1 differentiation (anti-human CD28 [0.5 µg/ml; #302933, BioLegend, San Diego, CA], IL2 [100 U/ml], IL12 [50 ng/µl], and anti-IL4 [20 ng/µl; #130 097 742 | #130-096-704 | # 130-095-709, Miltenyi Biotec, Bergisch Gladbach, Germany]) and for Th2 differentiation [anti-human CD28 (0.5 µg/ml), IL2 (100 U/ml), IL4 (50 ng/ml), anti-IFNγ (50 ng/ml), and anti-IL12 (50 ng/ml; #130 097 742 | #130-096-753 | #130 095-743 | # 130-103-738, Miltenyi Biotec, Bergisch Gladbach, Germany)], respectively. After 4 days, the cell suspensions were transferred to an uncovered cell culture plate and incubated for another 2 days with the same supplements except anti-human CD28.

### $T_{mem}$ differentiation

To differentiate memory T cells ($T_{mem}$) *in vitro*, PBMCs were incubated and T cells specifically stimulated as described above. After 48 h, CD3/CD28 Dynabeads were magnetically removed and CD4[+]- and CD8[+]-cell isolation was performed. Isolated cells were then incubated with 50 U/ml IL2 and 25 ng/ml IL15 for an additional 3 days.

### Cytotoxicity assay

Cytolytic T-cell function was evaluated using a cytotoxicity assay. Isolated T cells were co-cultivated with calcein-labeled U87 glioblastoma cells (8 µM calcein AM; #C1359, Sigma-Aldrich, Darmstadt, Germany). Lysis-dependent calcein fluorescence was measured on a Filtermax F3 (Molecular devices. LLC, San Jose, CA, USA). Fluorescence values were obtained using a 480 nm excitation filter and a 520 nm emission filter.

### Phagocytosis assay

Innate immune cell phagocytic activity was measured using the Green E. coli Phagocytosis Assay Kit (# PK-CA577-K963, PromoCell, Heidelberg, Germany) according to the manufacturer's instructions. Cellular phagocytic activity was quantified as mean fluorescence intensity (MFI) green using a FACS Canto II flow cytometer (BD Biosciences, Franklin Lakes, NJ, USA).

### Whole blood multiplex protein analysis

For multiplex protein analyses, blood samples were taken from healthy volunteers prior to the start (T0) and on the last day of the KD (T1), using *truculture* tubes—containing medium and lipopolysaccharide (LPS)—according to the manufacturer's protocol. Multiplex analysis (HumanCytokineMAP® A/B) was performed by myriadRBM (Austin, Texas, USA).

### ELISA

Levels of secreted proteins were quantified by enzyme-linked immunosorbent assay (ELISA; IFNγ: #430104; IL2: #431804, IL4: #430304, IL6: #430504, IL8: #431504, IL10: #430604, IL1β: #437004, TNFα: #430204; BioLegend). Assays were performed according to the manufacturer's protocol. Absorbance was measured on a Filtermax F3 and values evaluated by using a plate-specific standard curve.

### Oxygen consumption rate and extracellular acidification rate

Mitochondrial function was analyzed by extracellular flux analysis using a Seahorse XF96 Analyzer (Agilent, Santa Clara, USA). CD4[+] and CD8[+] T cells were plated on a poly-L-lysine-coated (Biochrom,

# L7240, Berlin, Germany) 96-well plates (#102601-100, Agilent, Santa Clara, USA) in Assay Medium (containing Seahorse RPMI supplemented with 1 mM sodium pyruvate, 2 mM glutamine and 5.5 mM glucose for the Mito Stress Test or supplemented with 5 mM HEPES for the Glycolytic Rate Assay). All experiments were performed in triplicates using 200 000 cells per well. To measure extracellular acidification rate (ECAR), oxygen consumption rate (OCR) and glycolytic proton efflux rate (glycoPER) Mito Stress Test (#103015-100) and Glycolytic Rate Assay (# 103344-100) was performed. To analyze mitochondrial oxidative phosphorylation, final well concentrations of 1 μM Oligomycin, 0.75 μM FCCP and 0.5 μM Rotenone/Antimycin A were consecutively added through designated Seahorse cartridge compound delivery ports. For the evaluation of cellular glycolysis, 0.5 μM Rotenone/Antimycin A and 50 mM 2-deoxy-glucose were injected accordingly.

## Mitochondrial membrane potential

The membrane permeable lipophilic cationic carbocyanine dye JC1 was used to evaluate mitochondrial membrane potential $\Delta\psi M$ as per the manufacturer's instructions (Item No. 701560, Cayman Chemical, Ann Arbor, MI, USA) and analyzed on a FACS Canto II flow cytometer (BD Biosciences, Franklin Lakes, NJ, USA). JC1 accumulates $\Delta\psi M$-dependent in intact mitochondria, forming J aggregates and emitting red fluorescence (~ 590 nm). Lower cellular dye concentrations due to depolarized mitochondrial membrane potential results in the accumulation of green fluorescent monomeric forms of JC1 (~ 529 nm). $\Delta\psi M$ is depicted as the mean fluorescence intensity ratio of red/green. FCCP (carbonyl cyanide-$p$-trifluoromethoxyphenylhydrazone), an uncoupling ionophoric agent causing complete loss of $\Delta\psi M$, served as a negative control.

## Cellular reactive oxygen species

Intracellular levels of reactive oxygen species (ROS) were quantified using CellROX Green Flow Cytometry Assay Kit (C10492, Thermo Fisher Scientific, Waltham, MA, USA) as per the manufacturer's protocol. $N$-acetylcysteine (NAC) and tert-butyl hydroxyperoxide (TBHP) were applied as negative and positive control, respectively.

## Mitochondrial superoxide production

MitoSOX Red mitochondrial superoxide indicator was utilized for the detection of mitochondrial ROS (M36008, Thermo Fisher Scientific, Waltham, MA, USA). The dye is selectively oxidized by respiratory chain-derived superoxides, emitting red fluorescence (~ 580 nm). Cells were stained with 0.2 μM MitoSOX in prewarmed hanks balanced salt solution (HBSS), incubated in the dark at 37°C and immediately analyzed using a FACS Canto II flow cytometer (BD Biosciences, Franklin Lakes, NJ, USA).

## Cellular glutathione content

To assess the intracellular level of glutathione (GSH), ThiolTracker (T10095, Thermo Fisher Scientific, Waltham, MA, USA) was used as per the manufacturer's protocol. Analysis was performed using a FACS Canto II flow cytometer (BD Biosciences, Franklin Lakes, NJ, USA).

## Mitochondrial mass determination

MitoTracker Green FM (#9074; Cell Signaling Technology, Danvers, MA, USA) was used to determine the mitochondrial mass via flow cytometry analysis. Cells were incubated with 200 nM MitoTracker in the dark at 37°C for a duration of 15 min. Mitochondrial mass per cell was subsequently determined by quantification of mean fluorescence intensity (MFI) green using a FACS Canto II flow cytometer (BD Biosciences, Franklin Lakes, NJ, USA).

For microscopic imaging, CD4$^+$ and CD8$^+$ T cells were incubated as described above. After two additional washing steps, lymphocytes were seeded in μ-slides 2 × 9 well (Ibidi, Gräfelfing, Germany). Images of 10 randomly chosen microscopic fields per well at 200× magnification and 3× digital zoom were acquired on a LEICA-TCS SP5 Confocal Microscope (Leica, Wetzlar, Germany) using Leica application suite AF software, version 2.7.

## Immunoblot

Cells were lysed in cell lysis buffer (#9803; Cell Signaling Technology Inc., Frankfurt am Main, DE) supplemented with a protease and phosphatase inhibitor (#5871 | #5870, Cell Signaling Technology Inc., Frankfurt am Main, DE). Equivalent protein concentrations were loaded onto polyacrylamide stacking gels. After transfer, protein was blotted with 1:400 Total OXPHOS Human Antibody Cocktail (#ab110411; Abcam, Cambridge, UK). As a loading control, 1:1,000 beta-Actin antibody (#5125, Cell Signaling Technology Inc., Frankfurt am Main, DE) was used.

## Flow cytometry

For CD3/CD4/CD8a antibody staining, $2 × 10^5$ cells were pre-incubated with 2.5 μl Human TruStain FcX™ Fc Receptor Blocking Solution (#422302, BioLegend, San Diego, CA, USA), followed by incubation with the designated antibody (Brilliant Violet 421™ anti-human CD3 Antibody (#317344), PerCP anti-human CD4 (#300528), Brilliant Violet 421™ anti-human CD8a (#301036), and the respective isotype controls (BioLegend, San Diego, CA, USA) for 30 min on ice while being protected from light.

Intracellular assessment of interferon γ and granzyme B has been performed using FITC anti-human interferon γ (#502506, BioLegend, San Diego, CA, USA) and BV421 anti-human granzyme B (#396413, BioLegend, San Diego, CA, USA) after fixation and permeabilization using eBioscience™ Fixation/Permeabilization Concentrate, Diluent and Buffer (#00-5123-43 | #00-5223-56 | #00-8333-56, Invitrogen, Carlsbad, USA) according to the manufacturer's instructions.

For $T_{reg}$ analyses, CD4$^+$ T cells were stained with CD25-PE and Foxp3-Alexa Fluor 488 (#302605 | #320111, Biolegend, San Diego, CA, USA). After initial surface staining (CD25) as described above, for fixation and permeabilization the eBioscience™ Foxp3 transcription factor Staining Buffer Set (# 00-5523-00, Thermo Fisher Scientific, Waltham, MA, USA) was used, followed by intracellular staining at 4°C for a duration of 25 min.

For assessing $Th_1$ and $Th_2$ T helper cell subsets, CD4$^+$ T cells were stained with PE anti-human CXCR3 (#353705, BioLegend, San Diego, CA, USA), Brilliant Violet 421 anti-human CCR4 (#359413, BioLegend, San Diego, CA, USA), FITC anti-human CCR6 (#353411, BioLegend, San Diego, CA, USA), and the respective isotype

controls. Th$_1$ T cells were defined as CCR6$^-$/CXCR3$^+$/CCR4$^{-/+}$, Th$_2$ as CCR6$^-$/CXCR3$^-$/CCR4$^+$.

T-cell proliferation was determined by fluorescent cell staining dye 5-(and 6-) carboxyfluorescein diacetate succinimidyl ester Kit (CFSE, Biolegend, San Diego, USA) according to the manufacturer's protocol.

To differentiate between memory T-cell subsets, MACS-isolated CD4$^+$ and CD8$^+$ T cells were stained with anti-human CCR7 PE, CD45RA PerCP, and CD45RO BV421 (all Biolegend: #353203, #304121, and #304223) as described above. Memory cells were identified as CD45RA$^-$/CD45RO$^+$. CCR7 was used to distinguish between central and effector memory cell subpopulations.

Flow cytometry data were acquired on a FACS Canto II (BD Biosciences, Franklin Lakes, NJ, USA). Data analyses were performed using FlowJo v10 (FlowJo, Ashland, USA).

### RNA isolation

Total RNA was isolated from primary T cells using the mirVana miRNA Isolation Kit (Thermo Fisher Scientific, Waltham, MA, USA) with subsequent DNase treatment (Turbo DNase, Thermo Fisher Scientific, Waltham, MA, USA) or miRNeasy RNA Isolation Kit (#217004, Qiagen, Hilden, Germany) with on-column DNA digestion. For RNA purification, the Monarch RNA Clean-up Kit (New England Biolabs, Ipswich, MA, USA) was utilized. All isolation procedures have been performed according to the respective manufacturer's instructions. RNA quantity and purity were measured using a NanoDrop 2000 spectrophotometer (Thermo Fisher Scientific, Waltham, MA, USA). First-strand cDNA was synthesized from equal amounts of RNA using SuperScript III reverse transcriptase (Invitrogen, Carlsbad, CA, USA), random hexamers, and oligo(dT) primers.

### Quantification of single gene mRNA expression

Expression of mRNA was determined using a LightCycler 480 instrument (Roche Diagnostics, Mannheim, Germany) as previously described (Möhnle et al, 2018; Hirschberger et al, 2019). All analyses were calculated using duplicate values. Succinate dehydrogenase subunit A (SDHA), β-actin, and TATA box-binding protein (TBP) served as reference genes in all experiments (Ledderose et al, 2011). Specifications of qPCR primer and probes are given in Table S1. Mean target/reference and target/reference standard deviation has been calculated for each mRNA target. Determination of quantification cycles has been performed by the LightCycler software using the second derivative maximum method. A quantification cycle (Cq) cut-off was defined for mRNA quantification (Cq 35). Cq values beyond cut-off were considered unspecific.

### Transcriptome profiling

The transcriptome of CD4$^+$ and CD8$^+$ human T cells before and after KD was determined by 3′-RNA sequencing. Sequencing libraries were generated using the QuantSeq 3′ mRNA-Seq Library Prep Kit FWD for Illumina (Lexogen GmbH, Austria). The optimal number of amplification cycles was determined using the PCR Add-on Kit for Illumina (Lexogen GmbH, Austria). Amplification was conducted according to the manufacturer's protocol. Quality and quantity of

**The paper explained**

**Problem**
The ketogenic diet (KD) is characterized by very limited uptake of carbohydrates resulting in endogenous production of ketone bodies as alternative energy substrates that can be utilized via mitochondrial aerobic oxidative phosphorylation There are as yet unproven assumptions that KD positively affects human immunity. We investigated this topic in an *in vitro* model using primary human immune cells and in a nutritional intervention study enrolling healthy volunteers.

**Results**
Ketogenic diet markedly improved specific responses of human T lymphocytes in a balanced way—including T effector and T regulatory cell function—and increased the formation of memory T cells both *in vitro* and *in vivo*. This effect was based on a redirection of T-cell metabolism toward aerobic mitochondrial oxidation, resulting in enhanced cellular energy supply and respiratory reserve. These functional changes were in line with transcriptomic alterations, linking the KD to a fundamental immunometabolic reprogramming of human T cells.

**Impact**
Our data suggest KD as a feasible and effective clinical tool to augment human T-cell immunity. This could impact various clinical issues intimately correlated to T-cell immune disorders. In conclusion, our study changes the perspective on nutrition as a clinical tool and could help to redefine the role of dietary interventions in modern medicine.

sequencing libraries were determined using the Quanti-iT PicoGreen dsDNA Assay Kit (Invitrogen, USA) and the Bioanalyzer High Sensitivity DNA Analysis Kit (Agilent Technologies, Inc., USA). Sequencing of libraries was performed on an Illumina HiSeq4000 sequencing machine (Illumina, Inc., USA). Individually barcoded libraries were pooled and distributed across lanes of the same flow-cell aiming for approximately 10 million paired-end reads per sample.

Raw forward reads as stored in fastq files were subjected to alignment against the human genome reference genome (hg38) using STAR after removing the adapter sequencing using BBDUk (https://jgi.doe.gov/data-and-tools/bbtools) (Dobin et al, 2013). Aligned reads were quantified using htseq-count (Anders et al, 2015). Quality of unaligned and aligned reads was assessed using FastQC (https://www.bioinformatics.babraham.ac.uk/projects/fastqc/), and the resulting reports were summarized using the multiQC tool (https://multiqc.info).

Only genes for which the number of reads for the whole data set was five times greater than the number of samples were kept. Plausibility of data was checked by assessing expression of the sex-specific XIST gene with an expected high expression in females compared to a low expression in males. Data consistency was analyzed by building a correlation heatmap which also allowed the detection of technical outliers. Differential expression analysis was performed using DESeq2 while statistical significance was accepted for corrected *P*-values (FDR) smaller than 25% (Love et al, 2014). Gene set enrichment analysis (GSEA) was conducted on all genes kept in the analysis ranked after shrinked (apeglm) log$_2$-fold-change using the command-line version of the GSEA software (version 4.0.3) (Subramanian et al, 2005). After quality control, 11,545 genes

remained in the data set. Correlation heatmap analysis identified the CD4-cell expression profiles of two patients as technical outliers. The final data set contained the CD4 expression profiles of 13 patients and the CD8 expression profiles of 15 patients.

### Statistical analysis

Statistical analysis was performed using GraphPad Prism 7.03 (GraphPad Software, Inc., USA). All healthy volunteers performed the KD; T cells were analyzed prior to start (T0) and at the end of the diet (T1). Thus, no blinding or randomization was performed. Paired *t*-test or Wilcoxon matched-pairs signed rank test, as appropriate, served for comparisons. Normal distribution was tested using D'Agostino & Pearson test and Shapiro–Wilk test. Data are depicted as box plots with median, 25th and 75th percentiles and range, if not stated otherwise. *P < 0.05, **P < 0.01, ***P < 0.001. Statistical parameters and biological replicates are reported in the figure legends. Statistical details for RNAseq analysis are specified in the respective methods section.

## Data and materials availability

The data sets produced in this study are available in the following database:

RNAseq analysis: Gene Expression Omnibus, accession number GSE158407, https://www.ncbi.nlm.nih.gov/geo/query/acc.cgi?acc = GSE158407.

**Expanded View** for this article is available online.

### Acknowledgements
This research was funded by institutional grants of the Ludwig-Maximilians-University (LMU) Munich. We thank Gabriele Gröger, Gudrun Prangenberg, Sandra Haßelt, and Jessica Drexler for their expert technical assistance. We thank http://istock.com/7romawka7 for providing pictogram license. Open Access funding enabled and organized by Projekt DEAL.

### Author contributions
Conceptualization: GS, SH, SK; Methodology: GS, SH, MH; Validation: GS and SH; Formal analysis: SH, GS, KU; Investigation: GS, SH, DE, XM, AF, JH, TW, TR, HM, MBM JH, KU; Resources: SH, GS, NE, and KU; Data curation: GS and SH; Writing original draft: SH, GS, KU, FWK, SK; Visualization: SH, KU, DE, GS; Project administration: SH and GS; Supervision: SK, SH, and GS contributed equally to this work. All authors discussed the results, commented on the manuscript, and agreed to the published version of the manuscript.

### Conflict of interest
The authors declare that they have no conflict of interest.

### For more information
https://www.wbex.med.uni-muenchen.de/pages/principal_s_kreth.htm

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
