## [Review Process File · EMBO Molecular Medicine]

Very Low Carbohydrate Diet Enhances Human T Cell Immunity through Immunometabolic Reprogramming

Simon Hirschberger, Gabriele Strauß, David Effinger, Xaver Marstaller, Alicia Ferstl, Martin Müller, Tingting Wu, Max Hübner, Tim Rahmel, Hannah Mascolo, Nicole Exner, Julia Hess, Friedrich Kreth, Kristian Unger, and Simone Kreth

DOI: [10.15252/emmm.202114323](https://doi.org/10.15252/emmm.202114323)

Corresponding author(s): Simone Kreth (simone.kreth@med.uni-muenchen.de), Simon Hirschberger (simon.hirschberger@med.uni-muenchen.de)

Review Timeline:

Submission Date:	24th Mar 21
Editorial Decision:	30th Apr 21
Revision Received:	20th May 21
Editorial Decision:	27th May 21
Revision Received:	1st Jun 21
Accepted:	2nd Jun 21

Editor: Zeljko Durdevic

Transaction Report:

30th Apr 2021

Dear Dr. Kreth,

Thank you for the submission of your manuscript to EMBO Molecular Medicine, and please accept my apologies for the delay in getting back to you. We have received feedback from two of the three reviewers who agreed to evaluate your manuscript. Should referee #1 provide a report, we will send it to you, with the understanding that we will not ask for an additional revision. As you will see from the reports below, the referees acknowledge the interest of the study but also raise important and partially overlapping concerns that should be addressed in a major revision.

We would welcome the submission of a revised version within three months for further consideration. However, we realize that the current situation is exceptional on the account of the COVID-19/SARS-CoV-2 pandemic. Please let us know if you require longer to complete the revision.

I look forward to receiving your revised manuscript.

Yours sincerely,

Zeljko Durdevic

***** Reviewer's comments *****

Referee #2 (Comments on Novelty/Model System for Author):

The authors present interesting findings. However, their in vitro data showing that BHB-induces an increase in ROS is inconsistent with findings in vivo showing that BHB reduces ROS. The author's in vivo data support their main conclusions, but parts of their in vitro data do not. They will need to address the discrepancies in their revised manuscript. It is unclear if the immunometabolic reprogramming seen in activated T cells in vitro, which involves aerobic glycolysis, is similar to the metabolic reprogramming that occurs for these cells in vivo, which involves OxPhos. The authors will need to address these discrepancies.

Referee #2 (Remarks for Author):

Summary

The authors present data suggesting that very low carbohydrate diets can improve human T cell immunity. In vitro and in vivo data are presented showing that BHB enhances CD4+, CD8+ and regulatory T cell capacity and augmented T memory cell formation. Their RNAseq and functional metabolic analyses showed that BHB-induced immunometabolic reprogramming favoring mitochondrial oxidative metabolism. The BHB-enhanced increase in oxygen consumption did not occur at the expense of glycolytic capacity in either activated primary T cells or in activated T cells that were isolated from human subjects treated with the KD. The author's results emphasize the value of nutrition and dietary interventions in modern medicine. While the in vivo data support the authors conclusions, there are several anomalies associated with the in vitro data that will require reevaluation.

Major Comments

1. The authors present evidence showing that the BHB-enhanced increase in oxygen consumption did not occur at the expense of glycolytic capacity in either activated primary T cells, or in activated T cells isolated from human subjects treated with the KD. The authors state that "Upregulation of mitochondrial oxidative phosphorylation was not on the cost of glycolysis but on top, as we observed unchanged glycolytic capacity of human KD T cells". It is known that immune cells, activated in vitro, will generate energy through aerobic fermentation (glycolysis) with elevated lactate production (doi: 10.1074/jbc.M114.551051; doi:10.1126/sciimmunol.aas9822). This is a unique feature of the in vitro environment that does not occur in activated T cells in vivo (doi:10.1126/sciimmunol.aas9822). T cells can elicit an immune response independent of glycolysis or aerobic fermentation. The authors should also recognize that oxygen consumption is not always a marker for oxidative phosphorylation, especially when using the Seahorse instrument in cultured cells (<https://doi.org/10.1016/j.isci.2020.101761>). Aerobic fermentation together with oxygen consumption, is also seen in many proliferating cells that are grown in vitro, but is not seen in cells grown in vivo, as recently described (<https://doi.org/10.1016/j.isci.2020.101761>). The authors will need to address these issues in citing references showing that OCR might not be indicative of oxidative phosphorylation when measured in cultured cells. They cannot therefore assume that OCR is synonymous with OxPhos. This should be mentioned.

2. The authors should consider the possibility that BHB might be increasing ATP synthesis through mitochondrial substrate level phosphorylation (mSLP) in the glutaminolysis pathway, as was recently presented (<https://doi.org/10.1016/j.isci.2020.101761>). The authors should also acknowledge this possibility.

3. It will be important for the authors to describe the BHB preparation they used in their experiments. Was this the D-BHB, the L-BHB, or a D/L racemic BHB? This is important, as only the D-BHB is produced naturally in vivo (doi:10.1016/j.plefa.2003.09.007).

4. The authors show that BHB increases ROS in the activated immune cells. This observation is not what is seen in intact tissue treated with D-BHB. The Veech group showed that D-BHB reduces ROS production by increasing the redox span of the CoQ couple (doi:10.1016/j.plefa.2003.09.007).

This increased redox span will reduce ROS production while increasing the efficiency of ATP hydrolysis (DOI 10.1002/iub.1627). The increased ROS seen in the in vitro-activated T cells is the result of the aerobic fermentation (Warburg effect) seen in these cells. Increased ROS is also seen in tumor cells, which use aerobic fermentation for ATP synthesis. The D'Agostino group showed that D-BHB will elevate ROS in tumor cells, but not in normal cells, which use OxPhos for ATP synthesis (DOI 10.1186/s12986-017-0178-2). The author's results in the activated T cells would be more in line with what is seen in tumor cells that also express aerobic fermentation than in normal cells that use OxPhos for ATP synthesis. The authors will need to address these issues, as the mechanisms by which BHB influences energy metabolism can be different between the in vitro and the in vivo environments.

5. The data in Table 1 show that BHB levels increased in blood during fasting. What was the effect of the fasting on blood glucose? The authors should discuss their findings in light of previous findings (PMCID: PMC2194504).

Referee #3 (Comments on Novelty/Model System for Author):

A dose response (in some of) the in vitro assays determine BHB effects on T cell cultures is recommended.

Referee #3 (Remarks for Author):

The paper by Hirschberger et al. deals with the effects of ketone bodies on human T lymphocytes, as well as with the effects of ketogenic diet in volunteers on T cell responses. Altogether, this is an interesting study, and the results are presented in a convincing and logical fashion. The authors might consider improving their paper in the following points:

Are the results obtained in Fig. 1 to 3 dose-dependent? The authors culture cells in the absence or presence (10 mM) of beta-hydroxybutyrate (BHB). What is the minimum concentration of BHB to obtain such effects? This appears important because ketogenic diet led to plasma BHB levels in the range of 1 to 2 mM, which is 5 to 10 times less than what the authors evaluated in vitro.

Would this perhaps explain why the in vitro results are somehow discrepant from the in vivo results. For example, in vitro TBET is downregulated by BHB while GATA3 is upregulated, while in vivo the ketogenic diet does not affect TBET but upregulates GATA3.

Fig.3A+B. Please indicate what was measured (CellROX dye and MitoSox) instead of FITC and PE. This remark applies to the rest of the figure where FITC measurements are indicated in an appropriate and inaccurate fashion.

Would it make sense to calculate ratios such as Treg cells among total CD4 T cells to understand the net effects of BHB and the ketogenic diet?

In the Introduction or in the Discussion, the authors should cite a recent paper by Ferrere G et al. (JCI Insight. 2021 Jan 25;6(2):145207) showing that ketogenic diet and ketone bodies enhance anticancer immune responses in mice. Indeed, the statement "Adaptive immunity, however, has not been addressed so far..." in the Introduction is not correct.

RESPONSE TO REVIEWERS

Referee #2

Major Comments

1. The authors present evidence showing that the BHB-enhanced increase in oxygen consumption did not occur at the expense of glycolytic capacity in either activated primary T cells, or in activated T cells isolated from human subjects treated with the KD. The authors state that "Upregulation of mitochondrial oxidative phosphorylation was not on the cost of glycolysis but on top, as we observed unchanged glycolytic capacity of human KD T cells". It is known that immune cells, activated in vitro, will generate energy through aerobic fermentation (glycolysis) with elevated lactate production (doi: 10.1074/jbc.M114.551051; doi:10.1126/sciimmunol.aas9822). This is a unique feature of the in vitro environment that does not occur in activated T cells in vivo (doi:10.1126/sciimmunol.aas9822). T cells can elicit an immune response independent of glycolysis or aerobic fermentation.

Glucose is commonly known as essential substrate for T cell function (Chapman et al, 2020; Peng et al, 2016), and multiple studies have emphasized the importance of aerobic glycolysis for activated T cells both in vitro and in vivo (Michalek et al, 2011; Macintyre et al, 2014; Chang et al, 2013; Buck et al, 2016). Moreover, T cells also gain energy by other pathways such as OXPHOS, glutaminolysis and the pentose phosphate pathway (Wang & Green, 2012). Major metabolic differences, however, are not detected in in vitro cultivated versus in vivo primary T cells, but occur upon activation and differentiation both in vitro and in vivo (Ma et al, 2019; Klarquist et al, 2018; Tan et al, 2017; Tarasenko et al, 2017). We now have discussed this issue on page 14, last paragraph.

The authors should also recognize that oxygen consumption is not always a marker for oxidative phosphorylation, especially when using the Seahorse instrument in cultured cells (<https://doi.org/10.1016/j.isci.2020.101761>). Aerobic fermentation together with oxygen consumption, is also seen in many proliferating cells that are grown in vitro, but is not seen in cells grown in vivo, as recently described (<https://doi.org/10.1016/j.isci.2020.101761>). The authors will need to address these issues in citing references showing that OCR might not be indicative of oxidative phosphorylation when measured in cultured cells. They cannot therefore assume that OCR is synonymous with OxPhos. This should be mentioned.

We agree that OCR measured via Seahorse is not necessarily indicative of OXPHOS which particularly concerns cultured cancer cells, which can cover parts of their energy demands by mitochondrial substrate level phosphorylation (mSLP) linked ATP production through glutaminolysis (Ramanathan et al, 2005; Seyfried et al, 2017). The resulting succinate,

however, is not further processed in the TCA cycle but secreted due to a reduced Succinate Dehydrogenase (SDH) activity thus leading to ATP synthesis without increasing oxygen consumption (Seyfried et al, 2017, 2020). Conversely, T cells are capable of oxidatively metabolizing succinate thus producing further ATP by OXPHOS. Glutamine-derived ATP generated via oxidative phosphorylation (including mSLP) can be detected via OCR. Only mSLP-derived ATP generated during fermentation of glutamine is masked in Seahorse analysis (Seyfried, 2012). Thus, in the context of this study - using primary human T cells - the potential error in estimating ATP through Seahorse analysis can be considered far lower than compared to tumor cell culture. However, detailed analysis of ATP delivering pathways affected by KD requires further investigation. **As suggested by the reviewer, we have added these important considerations to our manuscript on page 14, last paragraph and page 15, first paragraph.**

2. The authors should consider the possibility that BHB might be increasing ATP synthesis through mitochondrial substrate level phosphorylation (mSLP) in the glutaminolysis pathway, as was recently presented (<https://doi.org/10.1016/j.isci.2020.101761>). The authors should also acknowledge this possibility.

*Thank you for this interesting comment. Indeed, T cells are known to perform glutaminolysis and to rely on glutamine for immune function (Wang et al, 2011; Wang & Green, 2012). Upon T cell activation, complex changes to substrate utilization occur (Ma et al, 2019). We agree that BHB might impact this complex process, due to altered substrate availability, negative feedback and/or allosteric regulation. For example, the activity of glutamate dehydrogenase is controlled through sirt4, which is inhibited by fasting and low blood glucose, thereby increasing glutaminolysis (Li et al, 2011; Min et al, 2018). The precise effects of KD on glutaminolysis and mSLP in human KD T-cells, however, require further investigation. **As suggested, we now discuss this point in our manuscript on page 14, last paragraph and page 15, first paragraph.***

3. It will be important for the authors to describe the BHB preparation they used in their experiments. Was this the D-BHB, the L-BHB, or a D/L racemic BHB? This is important, as only the D-BHB is produced naturally in vivo (doi:10.1016/j.plefa.2003.09.007).

In our in vitro experiments, racemic D/L BHB has been used. As noted, only D-BHB is produced in vivo and is the only enantiomer that serves as a substrate for BHB dehydrogenase (Puchalska & Crawford, 2017). We performed in vitro dose finding experiments to determine the optimal BHB concentration for T cell culture (added to Appendix Figure S1). During a very short term BHB incubation of 48 hours, only 10mM (racemic) BHB displayed enhanced T cell immune capacity. In regard to these findings, we used 10mM D/L BHB in vitro, which is equivalent to approximately 5mM metabolically active D-BHB. This is in the range of endogenous levels of ketone bodies being maximally reached by a ketogenic diet, yet for only a very short incubation period without re-supplementation.

The use of racemic BHB and the dose finding experiments performed have been added to our manuscript (results, page 5, first paragraph | Appendix Figure S1).

4. The authors show that BHB increases ROS in the activated immune cells. This observation is not what is seen in intact tissue treated with D-BHB. The Veech group showed that D-BHB reduces ROS production by increasing the redox span of the CoQ couple (doi:10.1016/j.plefa.2003.09.007). This increased redox span will reduce ROS production while increasing the efficiency of ATP hydrolysis (DOI 10.1002/iub.1627). The increased ROS seen in the in vitro-activated T cells is the result of the aerobic fermentation (Warburg effect) seen in these cells. Increased ROS is also seen in tumor cells, which use aerobic fermentation for ATP synthesis. The D'Agostino group showed that D-BHB will elevate ROS in tumor cells, but not in normal cells, which use OxPhos for ATP synthesis (DOI 10.1186/s12986-017-0178-2). The author's results in the activated T cells would be more in line with what is seen in tumor cells that also express aerobic fermentation than in normal cells that use OxPhos for ATP synthesis. The authors will need to address these issues, as the mechanisms by which BHB influences energy metabolism can be different between the in vitro and the in vivo environments.

The role of ROS in T cell immunity is exceptional and markedly differs from tissues and particularly from cancer cells: ETC-derived ROS serve as a second messenger during T cell activation (Murphy & Siegel, 2013; Franchina et al, 2018), thus being considered pivotal for T cell immunity (Devadas et al, 2002; Jones et al, 2007; Sena et al, 2013). Of note, T cells diverge from NADPH-dependent GSH synthesis, instead redirecting NADPH for ROS

production via NADPH oxidase to fulfill their demand on reactive oxygen species (Jackson et al, 2004; Kwon et al, 2010). These findings gave rise to the concept of mitohormesis, opposing the idea of ROS as solely detrimental byproducts of an imperfect oxidative system, but emphasizing the role of ROS as essential signaling molecules (Ristow, 2014). We found a balanced and mild increase of mitochondrial ROS in response to KD in activated human T cells, both in vitro and in vivo, which did not compromise cellular anti-oxidative systems. This augmentation of T cell second messengers may thus be contributing to enhanced T cell immune capacity on a KD.

In other words, in metabolic therapy of cancer (as described by Seyfried et al.), the different effects of BHB-induced elevated ROS on tumor cells and T cells might even work in the same direction: While oxidative stress compromises cancer cell viability, mild increase of ROS enhances T cell immune capacity which in turn further restrains tumor growth.

We have discussed these interesting points on page 15, second paragraph and page 16, last paragraph.

5. The data in Table 1 show that BHB levels increased in blood during fasting. What was the effect of the fasting on blood glucose? The authors should discuss their findings in light of previous findings (PMCID: PMC2194504).

Fasting blood glucose concentrations were also measured and are now depicted in Table 1 and Figure EV3b, as requested. No significant changes in blood glucose concentrations could be detected, which is in line with previous data and probably the result of constant hepatic gluconeogenesis (Cahill & Veech, 2003). However, significant effects of KD on blood glucose are likely to occur over longer KD periods.

We have added these considerations to our manuscript (Table 1, Figure EV3b | discussion, page 14, second paragraph).

Referee #3

1. Are the results obtained in Fig. 1 to 3 dose-dependent? The authors culture cells in the absence or presence (10 mM) of beta-hydroxybutyrate (BHB). What is the minimum concentration of BHB to obtain such effects? This appears important because ketogenic diet led to plasma BHB levels in the range of 1 to 2 mM, which is 5 to 10 times less than what the authors evaluated in vitro.

Thank you for this important remark. As we used racemic BHB, 10mM refers to only 5mM metabolically active D-BHB. This concentration was used due to dose finding experiments, showing superior results in contrast to lower BHB concentrations. 5mM D-BHB are considered a near-maximum blood ketone body concentration in humans on a KD. Since our in vitro results cover only a very short time frame of 48 hours, this high concentration was chosen. In vivo, far lower concentrations over a far longer time period have been analyzed. Whether low concentrations of D-BHB in vitro, cultured over a longer time period may result in similar immunometabolic alterations requires further investigation.

We have added the dose finding experiments (flow cytometry and qRT-PCR) to our manuscript (results page 5, first paragraph / Appendix Figure S1).

2. Would this perhaps explain why the in vitro results are somehow discrepant from the in vivo results. For example, in vitro TBET is downregulated by BHB while GATA3 is upregulated, while in vivo the ketogenic diet does not affect TBET but upregulates GATA3.

In our dose finding experiments, no significant alteration of Tbet/GATA3 was detected in response to varying concentrations of BHB. Of note, the ratio of Tbet/Gata3 mRNA expression displays a similar decline in BHB/NC in vitro and T1/T0 in vivo, resulting in an overweight of Th2 transcription factor expression under both conditions. The observed differences in isolated mRNA expression levels thus might rather be due to different time periods in vivo and in vitro.

We have added this finding to our manuscript (results page 5, first paragraph, and page 11, first paragraph / Appendix Figure S1 + Appendix Figure S2).

3. Fig.3A+B. Please indicate what was measured (CellROX dye and MitoSox) instead of FITC and PE. This remark applies to the rest of the figure where FITC measurements are indicated in an appropriate and inaccurate fashion.

Thank you for this important remark.

As suggested, we have replaced FITC/ PE labeling with CellROX/ MitoSOX/ JCI/ ThioTracker throughout the manuscript to clearly indicate the measurement and to improve the clarity of the figures.

4. Would it make sense to calculate ratios such as Treg cells among total CD4 T cells to understand the net effects of BHB and the ketogenic diet?

We agree that the net effect of ketone bodies is best visible if regulatory T cells are calculated in regard to total CD4. Of note, in vivo Treg abundance is comparable to other studies (Churlaud et al, 2015; Rodríguez-Perea et al, 2015; Rueda et al, 2013).

Flow cytometry data of in vitro differentiated Treg (Figure 1) and in vivo Treg abundance (Figure 5) are depicted as % CD4+CD25+FoxP3+ of total CD4+ cells. Axis labeling and Figure captions have been updated accordingly.

5. In the Introduction or in the Discussion, the authors should cite a recent paper by Ferrere G et al. (JCI Insight. 2021 Jan 25;6(2):145207) showing that ketogenic diet and ketone bodies enhance anticancer immune responses in mice. Indeed, the statement " Adaptive immunity, however, has not been addressed so far..." in the Introduction is not correct.

We appreciate this important paper, as it consolidates the emerging evidence that KD impacts on T-cell immunity.

We have added the reference to our discussion (page 16, last paragraph) and we have updated the respective part of our introduction (page 3, second paragraph) as requested.

References

- Buck MD, O'Sullivan D, Klein Geltink RI, Curtis JD, Chang C-H, Sanin DE, Qiu J, Kretz O, Braas D, van der Windt GJW, *et al* (2016) Mitochondrial Dynamics Controls T Cell Fate through Metabolic Programming. *Cell* 166: 63–76
- Cahill GF Jr & Veech RL (2003) Ketoacids? Good medicine? *Trans Am Clin Climatol Assoc* 114: 149–61; discussion 162–3
- Chang C-H, Curtis JD, Maggi LB Jr, Faubert B, Villarino AV, O'Sullivan D, Huang SC-C, van der Windt GJW, Blagih J, Qiu J, *et al* (2013) Posttranscriptional control of T cell effector function by aerobic glycolysis. *Cell* 153: 1239–1251
- Chapman NM, Boothby MR & Chi H (2020) Metabolic coordination of T cell quiescence and activation. *Nat Rev Immunol* 20: 55–70
- Churlaud G, Pitoiset F, Jebbawi F, Lorenzon R, Bellier B, Rosenzweig M & Klatzmann D (2015) Human and Mouse CD8(+)CD25(+)FOXP3(+) Regulatory T Cells at Steady State and during Interleukin-2 Therapy. *Front Immunol* 6: 171
- Devadas S, Zaritskaya L, Rhee SG, Oberley L & Williams MS (2002) Discrete generation of superoxide and hydrogen peroxide by T cell receptor stimulation: selective regulation of mitogen-activated protein kinase activation and fas ligand expression. *J Exp Med* 195: 59–70
- Franchina DG, Dostert C & Brenner D (2018) Reactive Oxygen Species: Involvement in T Cell Signaling and Metabolism. *Trends in Immunology* 39: 489–502 doi:10.1016/j.it.2018.01.005 [PREPRINT]
- Jackson SH, Devadas S, Kwon J, Pinto LA & Williams MS (2004) T cells express a phagocyte-type NADPH oxidase that is activated after T cell receptor stimulation. *Nature Immunology* 5: 818–827 doi:10.1038/ni1096 [PREPRINT]
- Jones RG, Bui T, White C, Madesh M, Krawczyk CM, Lindsten T, Hawkins BJ, Kubek S, Frauwirth KA, Wang YL, *et al* (2007) The proapoptotic factors Bax and Bak regulate T Cell proliferation through control of endoplasmic reticulum Ca(2+) homeostasis. *Immunity* 27: 268–280
- Klarquist J, Chitrakar A, Pennock ND, Kilgore AM, Blain T, Zheng C, Danhorn T, Walton K, Jiang L, Sun J, *et al* (2018) Clonal expansion of vaccine-elicited T cells is independent of aerobic glycolysis. *Sci Immunol* 3
- Kwon J, Shatynski KE, Chen H, Morand S, de Deken X, Miot F, Leto TL & Williams MS (2010) The Nonphagocytic NADPH Oxidase Duox1 Mediates a Positive Feedback Loop During T Cell Receptor Signaling. *Science Signaling* 3: ra59–ra59 doi:10.1126/scisignal.2000976 [PREPRINT]
- Li M, Li C, Allen A, Stanley CA & Smith TJ (2011) The structure and allosteric regulation of glutamate dehydrogenase. *Neurochem Int* 59: 445–455
- Macintyre AN, Gerriets VA, Nichols AG, Michalek RD, Rudolph MC, Deoliveira D, Anderson SM, Abel ED, Chen BJ, Hale LP, *et al* (2014) The glucose transporter Glut1 is selectively essential for CD4 T cell activation and effector function. *Cell Metab* 20: 61–72

- Ma EH, Verway MJ, Johnson RM, Roy DG, Steadman M, Hayes S, Williams KS, Sheldon RD, Samborska B, Kosinski PA, *et al* (2019) Metabolic Profiling Using Stable Isotope Tracing Reveals Distinct Patterns of Glucose Utilization by Physiologically Activated CD8 T Cells. *Immunity* 51: 856–870.e5
- Michalek RD, Gerriets VA, Jacobs SR, Macintyre AN, MacIver NJ, Mason EF, Sullivan SA, Nichols AG & Rathmell JC (2011) Cutting edge: distinct glycolytic and lipid oxidative metabolic programs are essential for effector and regulatory CD4+ T cell subsets. *J Immunol* 186: 3299–3303
- Min Z, Gao J & Yu Y (2018) The Roles of Mitochondrial SIRT4 in Cellular Metabolism. *Front Endocrinol* 9: 783
- Murphy MP & Siegel RM (2013) Mitochondrial ROS fire up T cell activation. *Immunity* 38: 201–202
- Peng M, Yin N, Chhangawala S, Xu K, Leslie CS & Li MO (2016) Aerobic glycolysis promotes T helper 1 cell differentiation through an epigenetic mechanism. *Science* 354: 481–484 doi:10.1126/science.aaf6284 [PREPRINT]
- Puchalska P & Crawford PA (2017) Multi-dimensional Roles of Ketone Bodies in Fuel Metabolism, Signaling, and Therapeutics. *Cell Metab* 25: 262–284
- Ramanathan A, Wang C & Schreiber SL (2005) Perturbational profiling of a cell-line model of tumorigenesis by using metabolic measurements. *Proc Natl Acad Sci U S A* 102: 5992–5997
- Ristow M (2014) Unraveling the truth about antioxidants: mitohormesis explains ROS-induced health benefits. *Nat Med* 20: 709–711
- Rodríguez-Perea AL, Montoya CJ, Olek S, Chougnet CA & Velilla PA (2015) Statins increase the frequency of circulating CD4+ FOXP3+ regulatory T cells in healthy individuals. *J Immunol Res* 2015: 762506
- Rueda CM, Velilla PA, Chougnet CA & Rugeles MT (2013) Incomplete normalization of regulatory t-cell frequency in the gut mucosa of Colombian HIV-infected patients receiving long-term antiretroviral treatment. *PLoS One* 8: e71062
- Sena LA, Li S, Jairaman A, Prakriya M, Ezponda T, Hildeman DA, Wang C-R, Schumacker PT, Licht JD, Perlman H, *et al* (2013) Mitochondria are required for antigen-specific T cell activation through reactive oxygen species signaling. *Immunity* 38: 225–236
- Seyfried T (2012) *Cancer as a Metabolic Disease: On the Origin, Management, and Prevention of Cancer* John Wiley & Sons
- Seyfried TN, Arismendi-Morillo G, Mukherjee P & Chinopoulos C (2020) On the Origin of ATP Synthesis in Cancer. *iScience* 23: 101761
- Seyfried TN, Yu G, Maroon JC & D'Agostino DP (2017) Press-pulse: a novel therapeutic strategy for the metabolic management of cancer. *Nutr Metab* 14: 19
- Tan H, Yang K, Li Y, Shaw TI, Wang Y, Blanco DB, Wang X, Cho J-H, Wang H, Rankin S, *et al* (2017) Integrative Proteomics and Phosphoproteomics Profiling Reveals Dynamic Signaling Networks and Bioenergetics Pathways Underlying T Cell Activation. *Immunity* 46: 488–503
- Tarasenko TN, Pacheco SE, Koenig MK, Gomez-Rodriguez J, Kapnick SM, Diaz F, Zervas

PM, Barca E, Sudderth J, DeBerardinis RJ, *et al* (2017) Cytochrome c Oxidase Activity Is a Metabolic Checkpoint that Regulates Cell Fate Decisions During T Cell Activation and Differentiation. *Cell Metab* 25: 1254–1268.e7

Wang R, Dillon CP, Shi LZ, Milasta S, Carter R, Finkelstein D, McCormick LL, Fitzgerald P, Chi H, Munger J, *et al* (2011) The transcription factor Myc controls metabolic reprogramming upon T lymphocyte activation. *Immunity* 35: 871–882

Wang R & Green DR (2012) Metabolic reprogramming and metabolic dependency in T cells. *Immunological Reviews* 249: 14–26 doi:10.1111/j.1600-065x.2012.01155.x [PREPRINT]

27th May 2021

Dear Prof. Kreth,

Thank you for the submission of your revised manuscript to EMBO Molecular Medicine. I am pleased to inform you that we will be able to accept your manuscript pending the following final amendments:

- 1) Figures: The number of datapoints plotted in the panels appears to differ from the 'n' in the legend in Figures 1A-G, 2B/C, 3D/F/J/K, 5A/C-F, 6C/D/G/H, EV1A, EV2A/B/G, EV4F/H-J/L/M. Please revise and correct.
- 2) In the main manuscript file, please do the following:
 - Correct/answer the track changes suggested by our data editors by working from the attached/uploaded document.
 - Remove font colour.
 - Fig EV4I and J are called out as Fig EVI and J, please correct.
 - In M&M, provide the antibody dilutions that were used for each antibody.
 - In M&M, a statistical paragraph that should reflect all information that you have filled in the Authors Checklist, especially regarding randomization, blinding, replication.
 - Please merge "Funding" section with "Acknowledgements".
 - Add author contributions for Tingting Wu.
 - Rename "Competing interest" to "Conflict of interest".
 - Move "Author contributions", "Acknowledgement", "Conflict of interest" and "Data availability" sections after Materials and Methods.
 - Move Table 1, main Figure and EV Figure legends to the end of the manuscript. Remove Appendix table of content.
 - Please use the following format to report the accession number of your data:

The datasets produced in this study are available in the following databases:
[data type]: [full name of the resource] [accession number/identifier] ([doi or URL or identifiers.org/DATABASE:ACCESSION])

Please check "Author Guidelines" for more information.

<https://www.embopress.org/page/journal/17574684/authorguide#availabilityofpublishedmaterial>

- Funding: Please make sure that information about all sources of funding are complete in both our submission system and in the manuscript.

3) Appendix: Please remove font colour and correct figure and table nomenclature to "Appendix Figure S1" etc. and "Appendix Table S1", also in the main text.

4) Synopsis:

- Synopsis image: Please resize the synopsis image to 550 px-wide x (250-400)-px high and submit it as a high-resolution jpeg file.

- Please check your synopsis text and image, revise them if necessary and submit their final versions with your revised manuscript. Please be aware that in the proof stage minor corrections only are allowed (e.g., typos).

5) For more information: There is space at the end of each article to list relevant web links for further consultation by our readers. Could you identify some relevant ones and provide such information as well? Some examples are patient associations, relevant databases, OMIM/proteins/genes links, author's websites, etc...

6) As part of the EMBO Publications transparent editorial process initiative (see our Editorial at <http://embomolmed.embopress.org/content/2/9/329>), EMBO Molecular Medicine will publish online a Review Process File (RPF) to accompany accepted manuscripts. This file will be published in conjunction with your paper and will include the anonymous referee reports, your point-by-point response and all pertinent correspondence relating to the manuscript. Let us know whether you agree with the publication of the RPF and as here, if you want to remove or not any figures from it prior to publication. Please note that the Authors checklist will be published at the end of the RPF.

7) Please provide a point-by-point letter INCLUDING my comments as well as the reviewer's reports and your detailed responses (as Word file).

I look forward to reading a new revised version of your manuscript as soon as possible.

Yours sincerely,

Zeljko Durdevic

***** Reviewer's comments *****

Referee #2 (Remarks for Author):

The authors have done a good job in addressing previous concerns.

The authors performed the requested editorial changes.

We are pleased to inform you that your manuscript is accepted for publication and is now being sent to our publisher to be included in the next available issue of EMBO Molecular Medicine.

Corresponding Author Name: Simone Kreth, Simon Hirschberger

Manuscript Number: EMM-2021-14323